# Quantifying Correlation for Time Series Modelling Strategy: Evidence from SOFTS Variants

## Abstract

Channel-dependent (CD) and channel-independent (CI) strategies are competing inductive biases in long-term time series forecasting. Empirical evidence indicates that CD strategies benefit from stronger inter-channel correlations, yet the correlation structures underlying this advantage remain unexplored. Consequently, increasingly sophisticated CI, CD, and hybrid architectures continue to be proposed without a clear understanding of the spatial structures under which each is preferable. This study takes the hybrid CD/CI Series-cOre Fused Time Series (SOFTS) model and pushes it to the extremes. We introduce the fully CD Channel Mixer SOFTS (C-SOFTS), which maximizes channel mixing, and the fully CI Identity SOFTS (I-SOFTS), which eliminates all channel interactions. To characterise the spatial structure under which each of the three variants is preferable, we define two dataset descriptors: the high-correlation fraction, the proportion of highly correlated channel pairs, and block separation, the degree of separation between channel clusters. We find that I-SOFTS consistently outperforms the SOFTS baseline on few-channel, low-correlation datasets, while C-SOFTS outperforms SOFTS on datasets with high block separation, or with high-correlation fraction and a few clusters. The hybrid proves optimal only when the high-correlation fraction and block separation are moderately low. Our findings show that the effectiveness of the CD C-SOFTS depends not only on inter-channel correlational strength alone, but also on the nature of the correlations. We argue that reporting such spatial characteristics alongside performance metrics may improve interpretability, enable fairer model comparisons, and help practitioners select appropriate inductive biases for a given dataset.

## 1 Introduction

A central challenge in time series forecasting is determining how to model cross-channel relationships. Channel-dependent (CD) approaches explicitly capture inter-channel interactions, while channel-independent (CI) approaches treat each channel as a separate univariate sequence. The literature has split along this axis, with studies advocating for either strategy (Chen et al., 2023; Nie et al., 2023). More recently, hybrid models have emerged as a third approach that merges the strengths of both (Chen et al., 2024; Han et al., 2024a).

Multiple empirical studies have demonstrated that CI models often outperform CD models across multiple benchmark datasets. They converge faster with less training data, are highly adaptable since each channel learns its own temporal patterns independently, are unlikely to overfit during training, and achieve significantly better performance regardless of the implementation used (Han et al., 2024b; Nie et al., 2023). CI models are also more robust to distribution drift and scale more easily as they reduce model complexity. CD models have also been shown to outperform CI models when inter-channel correlation is sufficiently high, owing to their greater informational capacity and ability to capture cross-channel dependencies (Chen et al., 2023; Han et al., 2024b; Montero-Manso & Hyndman, 2021). In fact, one explanation for the consistent underperformance of CD models is that benchmark datasets exhibit minimal inter-channel dependence and therefore fail to represent the real-world conditions in which these models thrive (Abdelmalak et al., 2026; Chen et al., 2023).

Qiu et al. (2024) observed that the CD Crossformer model gradually surpassed the CI PatchTST model as inter-channel correlation increased. Likewise, Li et al. (2025) reported comparable performance between CI and CD models on weakly correlated datasets, but found that as correlation increased, CI models exhibited higher error rates, whereas CD models remained stable. Consistent with these findings, Abdelmalak et al. (2026) showed that CD models significantly outperformed CI models on strongly correlated datasets. Finally, Tan et al. (2025) found that CD models often outperformed CI models as multivariate complexity increased, although this advantage was not consistent and could reverse as relationships became more complex and nuanced. Despite these findings, no study has attempted to characterize the correlational conditions under which each strategy becomes preferable, and there is no consensus on how to measure correlation. For instance, the Traffic dataset has been classified as having both low (Abdelmalak et al., 2026) and high (Li et al., 2025) correlation.

This ambiguity is compounded by how modern deep forecasting evaluates models. Architectures are largely ranked through aggregate benchmark performance, despite growing evidence that inductive bias, rather than architectural novelty, governs performance. Comprehensive empirical evaluations consistently show that no single model achieves state-of-the-art performance across all datasets (Li et al., 2025; Tan et al., 2025). Brigato et al. (2026) demonstrated that with careful hyperparameter tuning, different forecasting models achieve competitive performance. A model's average rank across benchmarks therefore conflates regimes in which its inductive bias is well-matched with those in which it is not, obscuring the conditions that actually determine performance.

In classical statistical forecasting, this issue has long been addressed by selecting models based on measurable properties of the data. For example, stationarity determines the applicability of ARIMA models, seasonal structure motivates SARIMA formulations, and cointegration between series motivates vector error-correction models over independent vector autoregressive specifications (Li & Law, 2024). These diagnostics provide interpretable criteria that connect inductive biases to dataset characteristics, and enable fair model comparisons. We argue that a similar perspective is needed for deep learning forecasting models. Given the strong influence of inter-channel relationships on the relative effectiveness of CI and CD approaches, characterizing the spatial correlation structure of datasets may provide diagnostics for selecting appropriate channel interaction strategies, rather than relying solely on aggregate benchmark rankings.

In this study, we contribute to the forecasting literature by introducing new architectures and characterizing the spatial structure of datasets for which each is preferable. We conduct a controlled variant study of the Series-cOre Fused Time Series forecaster (SOFTS) (Han et al., 2024a), a hybrid multivariate time series model whose architecture cleanly separates CI and CD components, allowing us to push it to CI and CD extremes. We derive two variants: Identity SOFTS (I-SOFTS), the CI version, which removes all channel interaction; and Channel Mixer SOFTS (C-SOFTS), the CD version, which maximizes global channel interaction in the spatial and frequency domains. To characterize the spatial structure for which each variant succeeds or fails relative to each other, we introduce two data descriptors: high-correlation fraction, which is the proportion of highly correlated channel pairs; and block separation, which is the degree of separation between channel clusters. We show that these descriptors explain the relative performance of the three SOFTS variants, enabling practitioners to select the most appropriate variant for the data rather than relying on aggregate benchmark rankings.

Our contributions are as follows:

1. We introduce I-SOFTS and C-SOFTS as extreme CI and CD variants of SOFTS, enabling controlled isolation of cross-channel modeling effects.

2. We define high-correlation fraction and block separation as interpretable descriptors of inter-channel correlation structure.

3. We demonstrate that the magnitude of inter-channel correlation alone does not determine the effectiveness of the CD variant, and that the nature of these correlations is equally important.

## 2 CI and CD Architectural Variants

The recognition that neither strategy universally dominates has motivated hybrid architectures that support controlled channel interaction (Liu et al., 2025; Ma et al., 2025; Chen et al., 2024). One recent example is SOFTS (Han et al., 2024a), which aggregates all channels into a shared global representation before redistributing it to each channel independently, thereby keeping cross-channel interactions explicit and architecturally isolated.

To isolate the effect of channel interaction strategy from architectural confounds, we introduce CI and CD variants of SOFTS by modifying it in opposite directions along the CI–CD spectrum. This approach ensures that any performance difference between the hybrid and its variants is attributable solely to the degree of channel interaction, not to differences in model capacity, normalization, or other architectural properties.

The core contribution of SOFTS is the STar Aggregate-Redistribute (STAR) module. It models channel interaction by appending a global representation of all channels to each channel. I-SOFTS converts the STAR module into an identity operation, returning each channel's representation unchanged. C-SOFTS retains the STAR module while replacing the CI feedforward network with a Channel Mixer module, which maximizes channel dependence in both the frequency and spatial domains.

### 2.1 SOFTS

The original SOFTS model established an efficient framework for multivariate time series forecasting through a centralized channel interaction system. Given historical values $X \in \mathbb{R}^{C \times L}$ with $C$ channels and lookback window $L$, SOFTS predicts future values $\hat{Y} \in \mathbb{R}^{C \times H}$, where $H$ denotes the number of future time steps. SOFTS employs four key components (Figure 1): Reversible instance normalization, which standardizes each channel to zero mean and unit variance, then reverses the transformation after prediction; series embedding that projects each channel's temporal sequence into a $d$-dimensional representation $S_0 \in \mathbb{R}^{C \times d}$ via linear projection; $N$ encoder layers with the STAR module for channel interaction, and a linear predictor mapping the final representation to forecasts.

### 2.1.1 STAR Module

The STAR module replaces distributed attention mechanisms with centralized aggregation. For the encoder layer $i$, STAR first compresses all channel embeddings into a global core representation $o_i \in \mathbb{R}^{d_{\text{core}}}$ through an MLP projection followed by stochastic pooling:

$$o_i = \text{Stoch\_Pool}\big(\text{MLP}_{\text{projection}}(S_{i-1})\big) \tag{1}$$

During training, stochastic pooling samples a feature per channel; during evaluation, a weighted average is used. The core is then concatenated with each channel's embeddings and fused via another MLP:

$$\begin{aligned} F_i &= \text{Concat}\big(S_{i-1}, \text{repeat}(o_i)\big) \\ S_i &= \text{MLP}_{\text{fusion}}(F_i) \end{aligned} \tag{2}$$

This centralized design achieves linear complexity $O(Cd)$ compared to the quadratic $O(C^2 d)$ for channel-wise attention, while improving robustness to noisy channels through global aggregation.

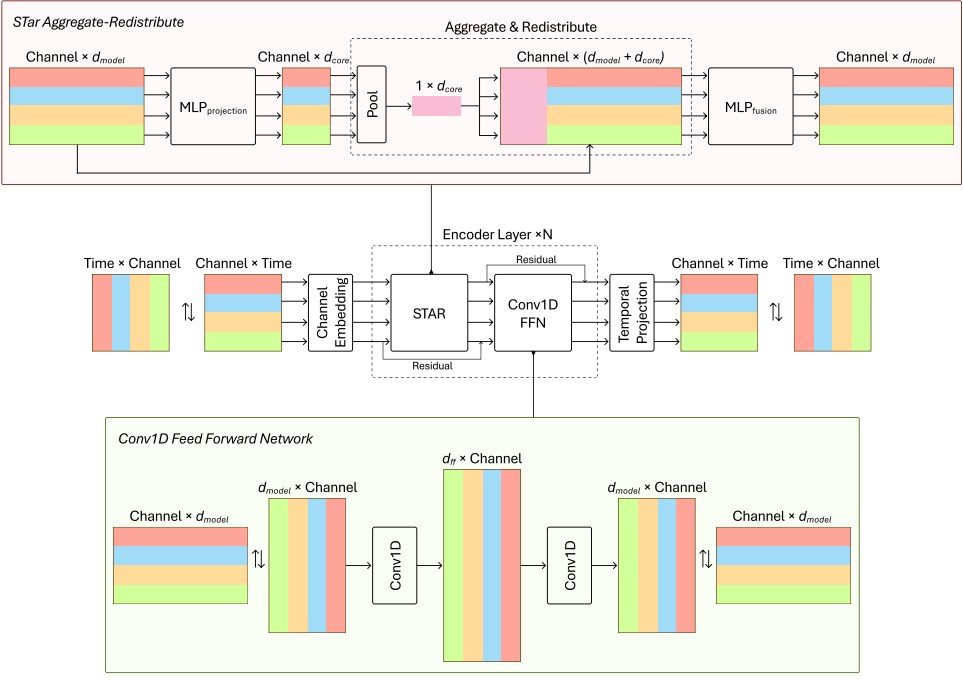

Figure 1: SOFTS for multivariate time series forecasting, introduced by Han et al. (2024a). Colored bars represent individual channels, and $N$ denotes the number of encoder layers. The input time series is first normalized and embedded into dimension $d$. The encoder comprises stacked layers that combine the STAR module with a position-wise feedforward network implemented using point-wise Conv1D. Channel dependencies are modeled in the STAR module, which aggregates channel features into a global representation that is then redistributed to each channel.

### 2.1.2 Encoder Layer

Each SOFTS encoder layer comprises a STAR block followed by a point-wise Conv1D feedforward network with residual connections and layer normalization. For input $x \in \mathbb{R}^{C \times d}$:

$$
\begin{aligned}
x' &= x + \text{Dropout}\big(\text{STAR}(x)\big) \\
x' &= \text{LayerNorm}(x') \\
y &= \text{Conv1D}(x'^{\top}, d \to d_{\text{ff}}) \\
y &= \text{GELU}(y) \\
y &= \text{Conv1D}(y, d_{\text{ff}} \to d)^{\top} \\
x'' &= x' + \text{Dropout}(y) \\
x_{\text{out}} &= \text{LayerNorm}(x'')
\end{aligned}
\tag{3}
$$

The two Conv1D layers act as a CI feedforward network, with hidden dimension $d_{\text{ff}}$.

## 2.2 Channel Mixer SOFTS (C-SOFTS)

C-SOFTS pushes SOFTS towards channel-dependence by replacing the CI Conv1D feedforward network with the Channel Mixer module that operates in the spatial and frequency domain along the channel dimension (Figure 2).

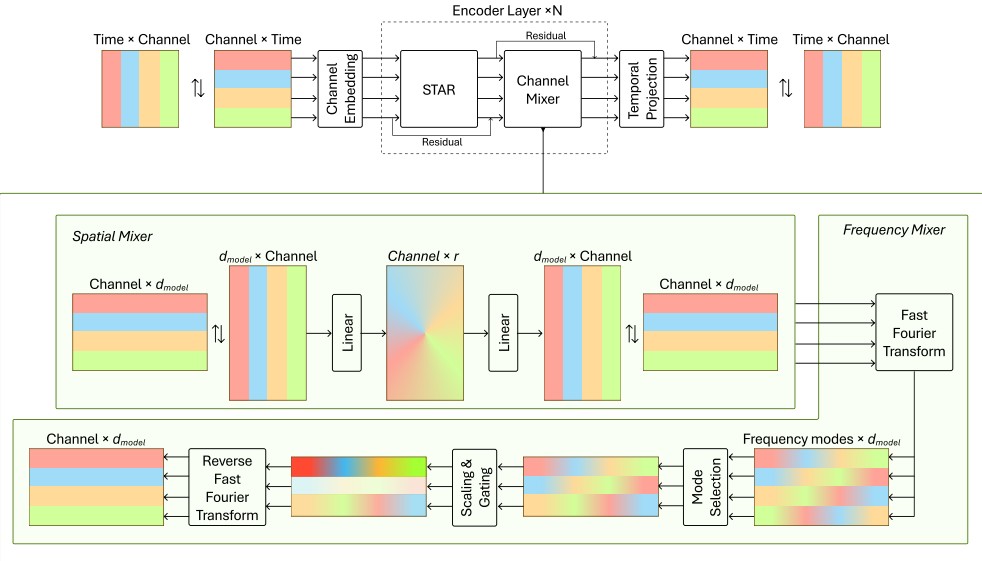

Figure 2: C-SOFTS is the CD variant of SOFTS. The Conv1D feedforward is replaced with the Channel Mixer module, which forces channel mixing in the spatial and frequency domains. The STAR module from SOFTS remains unchanged.

The Channel Mixer processes channel embeddings through four stages:

1. **Spatial Mixing:** A two-layer MLP applied along the channel dimension, which projects the input into a bottleneck dimension $r$ and reconstructs it, enabling channel mixing.

2. **Frequency-Domain Transformation:** The spatially mixed representation is transformed along the channel dimension using the Real Fast Fourier Transform (RFFT) with orthonormal normalization to preserve energy.

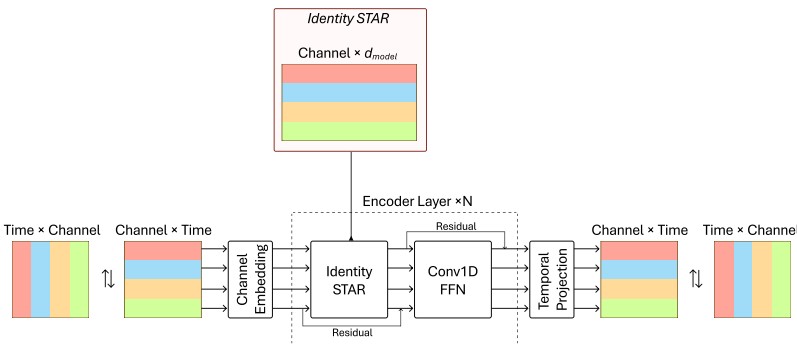

Figure 3: I-SOFTS is the CI variant of SOFTS. The STAR module returns the data unchanged, eliminating all cross-channel interaction. The Conv1D feedforward network from SOFTS remains unchanged.

3. **Learnable Frequency Filtering:** Each frequency mode is modulated by a learnable complex weight $W$ and a gating parameter $g = \sigma(G)$ to select important frequency filtering. Element-wise multiplication in the frequency domain is equivalent to a circular convolution across all channels in the spatial domain with a kernel length of $C$, enabling efficient cross-channel interactions with fewer parameters.

4. **Inverse Frequency-Domain Transformation:** The filtered frequency representation is mapped back to the spatial domain via the inverse RFFT.

In the encoder layer, the STAR block is now followed by the Channel Mixer:

$$\begin{aligned}
x' &= x + \text{Dropout}\big(\text{STAR}(x)\big) \\
x' &= \text{LayerNorm}(x') \\
x'' &= x' + \text{Dropout}\big(\text{ChannelMixer}(x')\big) \\
x_{\text{out}} &= \text{LayerNorm}(x'')
\end{aligned} \tag{4}$$

### 2.3 Identity SOFTS (I-SOFTS)

I-SOFTS is the extreme CI variant of SOFTS. It enforces strict CI by removing the global core representation $o_i$, and the projection and fusion MLPs in the STAR module, and replaces them with an identity mapping that returns the input representations unmodified (Figure 3):

$$\text{STAR}(S_{i-1}) = S_{i-1} \tag{5}$$

## 3 Dataset Descriptors

We characterize the spatial structure of datasets using descriptors derived from the channel-wise Pearson correlation matrix. These descriptors serve as pre-modeling diagnostics for selecting the appropriate SOFTS variant.

**High-correlation fraction** measures the proportion of channel pairs whose absolute Pearson correlation exceeds 0.5, a conventional threshold for moderate-to-strong correlation. High values indicate that a large proportion of channel pairs are strongly correlated, while low values indicate heterogeneous or weakly correlated channels.

153 **Block separation** measures how cleanly channels partition into internally coherent groups. It is defined as
154 the difference between the average within-cluster and between-cluster absolute Pearson correlation. Channel
155 clusters are identified using hierarchical clustering with Ward linkage applied to the correlation distance
156 matrix $D_{ij} = 1 - |r_{ij}|$, where $r_{ij}$ is the Pearson correlation between channels $i$ and $j$. The number of
157 clusters $k$ is selected over $k \in [2, \min(\lfloor C/2 \rfloor, 500) - 1]$ by maximizing the silhouette score computed on
158 the precomputed distance matrix. High values indicate strong block structure, where channels form tight,
159 internally coherent groups with weak cross-group interaction. Low values indicate diffuse correlation without
160 clear grouping.

## 4 Experiments

162 Forecast accuracy is evaluated using the Mean Squared Error (MSE) and Mean Absolute Error (MAE). The
163 performances of I-SOFTS and C-SOFTS are reported as the average percentage change in MSE relative to
164 the SOFTS baseline across all forecasting horizons. For each horizon, the percentage change is computed
165 with respect to SOFTS, and these values are then averaged. Positive values indicate improvement (reduced
166 MSE/MAE), whereas negative values indicate degradation (increased MSE/MAE).

167 Every single experimental setting (batch size, data split, learning rate, GPU, etc) was preserved exactly as
168 is in the SOFTS implementation[1], ensuring that any performance differences are attributable solely to the
169 architectural modifications in I-SOFTS and C-SOFTS.

### 4.1 C-SOFTS

171 Table 1 compares the performance of C-SOFTS and SOFTS, showing that C-SOFTS outperformed SOFTS
172 on six of the eight datasets.

Table 1: Results of C-SOFTS on eight benchmark datasets. The scores are averaged across horizons. The best results are bold and in red, and the second-best results are underlined and in blue. The results of SOFTS were reproduced for this study, and the other results are taken from the SOFTS paper (Han et al., 2024a)

| Models | C-SOFTS | | SOFTS | | iTransformer | | PatchTST | | TSMixer | | Crossformer | | DLinear | | SCINet | | FEDformer | |
|---|---|---|---|---|---|---|---|---|---|---|---|---|---|---|---|---|---|---|
| Metric | MSE | MAE | MSE | MAE | MSE | MAE | MSE | MAE | MSE | MAE | MSE | MAE | MSE | MAE | MSE | MAE | MSE | MAE |
| ECL | **0.168** | 0.268 | 0.176 | 0.266 | 0.178 | 0.270 | 0.189 | 0.276 | 0.189 | 0.276 | 0.244 | 0.334 | 0.212 | 0.300 | 0.268 | 0.365 | 0.214 | 0.327 |
| Traffic | 0.489 | 0.320 | **0.410** | **0.268** | 0.428 | 0.282 | 0.454 | 0.286 | 0.522 | 0.357 | 0.550 | 0.304 | 0.625 | 0.383 | 0.804 | 0.509 | 0.610 | 0.376 |
| Weather | **0.250** | **0.276** | 0.256 | 0.279 | 0.258 | 0.278 | 0.256 | 0.279 | 0.256 | 0.279 | 0.259 | 0.315 | 0.265 | 0.317 | 0.292 | 0.363 | 0.309 | 0.360 |
| Solar | 0.231 | 0.265 | **0.230** | **0.256** | 0.233 | 0.262 | 0.236 | 0.266 | 0.260 | 0.297 | 0.641 | 0.639 | 0.330 | 0.401 | 0.282 | 0.375 | 0.291 | 0.381 |
| PEMS03 | **0.097** | **0.197** | 0.107 | 0.212 | 0.113 | 0.221 | 0.137 | 0.240 | 0.119 | 0.233 | 0.169 | 0.281 | 0.278 | 0.375 | 0.114 | 0.224 | 0.213 | 0.327 |
| PEMS04 | **0.085** | **0.189** | 0.103 | 0.208 | 0.111 | 0.221 | 0.145 | 0.249 | 0.103 | 0.215 | 0.209 | 0.314 | 0.295 | 0.388 | 0.092 | 0.202 | 0.231 | 0.337 |
| PEMS07 | **0.088** | **0.170** | 0.088 | 0.184 | 0.101 | 0.204 | 0.144 | 0.233 | 0.112 | 0.217 | 0.235 | 0.315 | 0.329 | 0.395 | 0.119 | 0.234 | 0.165 | 0.283 |
| PEMS08 | **0.134** | **0.210** | 0.140 | 0.220 | 0.150 | 0.226 | 0.200 | 0.275 | 0.165 | 0.261 | 0.268 | 0.307 | 0.379 | 0.416 | 0.158 | 0.244 | 0.286 | 0.358 |

#### 4.1.1 Ablation Study

174 To evaluate the contribution of each component in the Channel Mixer, we conducted ablation experiments
175 by selectively removing the learned complex scaling weights, the mode gates, or the spatial mixer. All results
176 are reported relative to the full Channel Mixer, which serves as the baseline. Figure 4 shows the average
177 change in MSE across all forecasting horizons for each ablated component.

---

[1] https://github.com/Secilia-Cxy/SOFTS

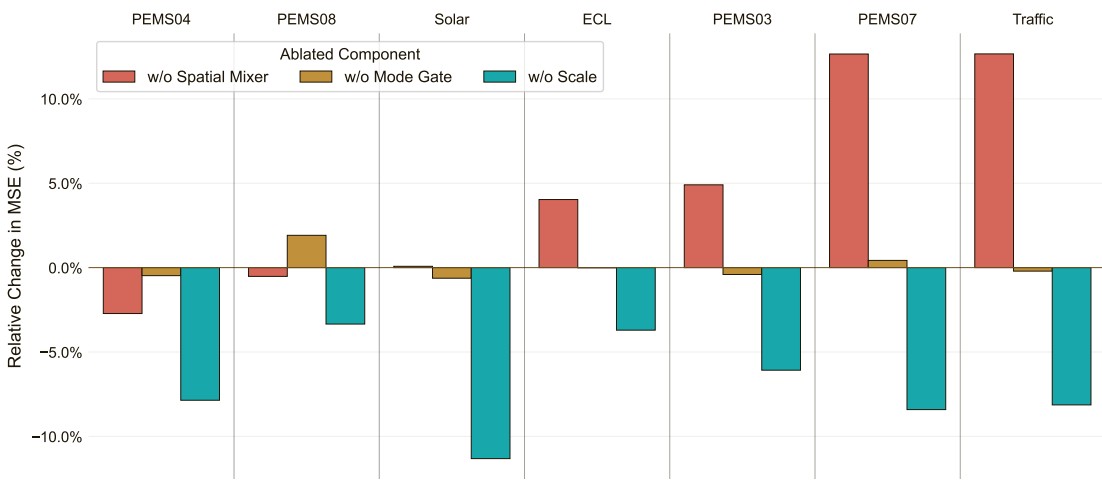

Figure 4: Impact of component ablation on average forecasting error. Negative values indicate that MSE worsens when a component is removed. All relative performances are computed using the complete Channel Mixer as the baseline. Complex scaling weights consistently improve performance, mode gates have minimal impact, and the spatial mixer's effect varies across datasets.

Removing the learned spectral scaling degrades performance on every dataset (-3.7% to -11.3%), with Solar showing the largest drop. This suggests that scaling via the complex weights is an essential mechanism behind C-SOFTS's CD approach. In contrast, ablating the mode gates has minimal impact (-0.6% to 0.4%) on all datasets except PEMS08, which improves MSE by 2.0%.

The impact of removing the spatial mixer varies substantially across datasets. On PEMS04 and PEMS08, its removal leads to MSE degradation of 2.7% and 0.5%, respectively, indicating that the spatial mixer is beneficial. In contrast, removing the spatial mixer yields notable improvements on other datasets, most prominently on PEMS07 and Traffic, where MSE increases by 12.7% in both cases. Beyond final accuracy, we observed that removing the spatial mixer consistently prolonged training convergence, often requiring almost twice the number of epochs.

These results show that while the complex scaling weights are universally beneficial, the spatial mixer's contribution varies across datasets.

### 4.1.2 Dataset Spatial Properties and C-SOFTS Performance

Table 2 presents the dataset properties alongside C-SOFTS's MSE performance relative to the SOFTS baseline, while Figure 5 shows the relative performance across forecasting horizons. C-SOFTS underperforms SOFTS on PEMS07, Solar, and Traffic, with the largest degradation on Traffic (-19.5%). It outperforms SOFTS on the remaining datasets, with consistent improvement across all horizons observed for Weather, ECL, and PEMS04. To explain why C-SOFTS degrades or improves over the hybrid, we group the datasets into three descriptive regimes based on the proposed dataset descriptors, in the context of the ablation study.

*Block Structure Regime*

In the block structure regime, channels organize into tight clusters with strong within-cluster cohesion and weak between-cluster correlation. Weather and ECL achieve MSE improvements of 3.0% and 4.3%, respectively, despite their relatively low high-correlation fraction, indicating that clear cluster separation alone is sufficient for C-SOFTS to outperform SOFTS. However, the ablation results show that these gains are achieved despite the degradation of the spatial mixer. For instance, on ECL, removing the spatial mixer improves C-SOFTS by a further 4.0%.

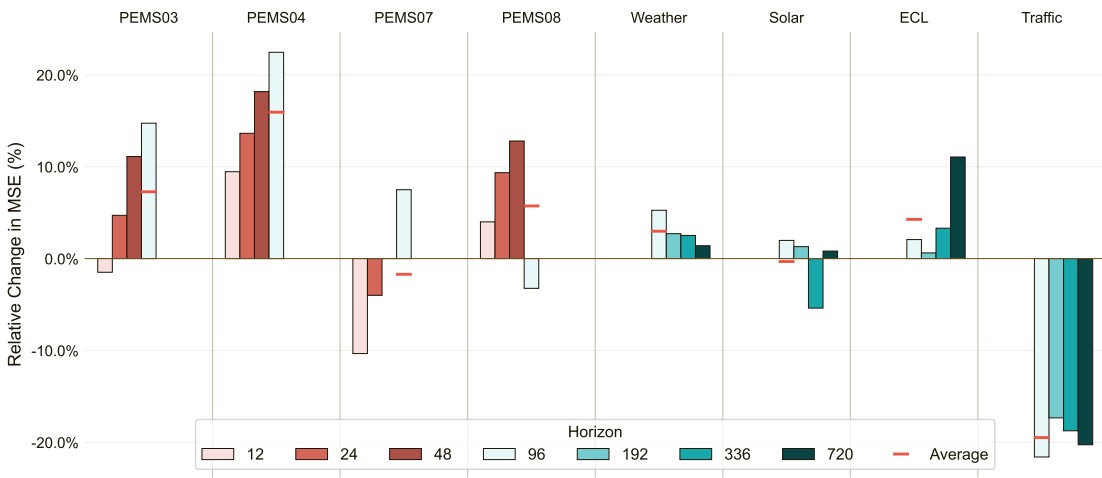

Figure 5: MSE performance of C-SOFTS over SOFTS across forecasting horizon. Bars show the relative change in MSE. Positive values favor C-SOFTS.

Table 2: Datasets' correlation structure and C-SOFTS performance relative to SOFTS.

| Dataset | Channels (# of clusters) | High-Correlation Fraction | Block Separation | Avg MSE Δ% | Spatial Correlation Regime |
|---------|--------------------------|---------------------------|------------------|------------|----------------------------|
| Weather | 21 (9) | 0.25 | 0.68 | 2.98 | Block Structure |
| ECL | 321 (4) | 0.46 | 0.46 | 4.27 | Block Structure |
| PEMS03 | 358 (99) | 1.00 | 0.13 | 7.27 | Homogeneous |
| PEMS04 | 307 (2) | 0.95 | 0.10 | 15.94 | Homogeneous |
| PEMS08 | 170 (3) | 0.96 | 0.12 | 5.16 | Homogeneous |
| PEMS07 | 883 (316) | 0.98 | 0.19 | -1.71 | Homogeneous |
| Solar | 137 (7) | 1.00 | 0.05 | -0.32 | Homogeneous |
| Traffic | 862 (2) | 0.67 | 0.18 | -19.49 | Incoherent |

*Homogeneous Regime*

Homogeneous regimes are characterized by uniformly high pairwise correlations with weak cluster separation. Within this category, performance depends on the degree to which meaningful spatial grouping exists and on whether the spatial mixer can exploit it constructively.

With a high-correlation fraction of 1 and a block separation score of 0.05, Solar's channels are essentially indistinguishable. There is no cluster structure to exploit beyond what the hybrid model captures. Even removing the spatial mixer does not meaningfully degrade performance. C-SOFTS neither gains nor loses meaningfully against the baseline (-0.32%).

The PEMS datasets demonstrate that block separation alone does not guarantee improvement and that high-correlation plays a significant role. PEMS04 and PEMS08 have very few clusters (2-3). Both the spatial and spectral frequency work constructively here. The spatial mixer compresses and reconstructs channels faithfully given their shared structure, and spectral modulation captures a dominant, consistent pattern across the dataset. Together, they yield strong improvements of 15.9% and 5.2%, respectively.

PEMS03 and PEMS07 have 99 and 316 clusters, respectively. In both cases, the spatial mixer is detrimental. Across the PEMS datasets, the utility of the spatial mixer decreases as the number of clusters increases. Ablating the spatial mixer degrades C-SOFTS performance on PEMS04 (2 clusters) and PEMS08 (3 clusters) by -2.7% and -0.5%, respectively. In contrast, performance improves on PEMS03 (99 clusters) and PEMS07 (316 clusters) by 4.9% and 12.7%, respectively. The difference in overall performance, particularly for PEMS03 and PEMS07, depends on whether the gains from spectral modulation are sufficient to overcome the degradation introduced by the spatial mixer. This is the case for PEMS03, which still achieves a 7.3% improvement over SOFTS, but not for PEMS07, where the degradation from the spatial mixer results in a net decline of 1.71%.

*Incoherent Regime*

Traffic exhibits the most severe degradation, with C-SOFTS performing 19.5% worse. Its high-correlation fraction of 0.67 and block separation of 0.18 place it in an intermediate regime. This level of correlation is neither sufficiently strong to provide a reliable inter-channel signal nor sufficiently structured to yield meaningful cluster contrast. With 862 channels collapsed into just 2 clusters, the spatial mixer aggregates weakly related signals rather than genuinely redundant ones. This distinguishes Traffic from PEMS04, which shares the same 2-cluster partition but has a high-correlation fraction of 0.95. Similar to PEMS07, ablating the spatial mixer improves performance by 12.7%. However, unlike PEMS07, where this recovery allows the model to outperform SOFTS, Traffic's performance deficit is too large to overcome. This case illustrates that additional global channel mixing offers no benefit over the hybrid approach.

In conclusion, the full C-SOFTS model improves over SOFTS when spatial structure is either homogeneous with few coherent clusters (PEMS04, PEMS08) or strongly block-structured (ECL, Weather), regardless of the degree of high correlation. However, when the data are homogeneous with many clusters (PEMS07), the spatial mixer becomes detrimental and should be removed. More broadly, it should only be used in settings with homogeneous data and few clusters. C-SOFTS offers no benefit when channels are nearly uniform with no exploitable cluster structure (Solar) or when correlations are moderate but lack clear spatial organization (Traffic).

## 4.2  I-SOFTS

Table 3 and Figure 6 show that, on average, I-SOFTS outperforms SOFTS across all datasets and horizons. ETTm2 improves across all horizons with an average MSE reduction of 1.74%. Other notable improvements occur at horizon 336 for ETTm1 (2.88%) and ETTh2 (2.77%). I-SOFTS maintains or improves MSE while eliminating STAR module computations, challenging the assumptions that balanced CI-CD hybrid strategies provide an optimal middle ground.

Table 3: Results of I-SOFTS on the four ETT datasets. The scores are averaged across horizons. The best results are bold and in red, and the second-best results are underlined and in blue. The results of SOFTS were reproduced for this study, and the other results are taken from the SOFTS paper (Han et al., 2024a) .

| Models | I-SOFTS | | SOFTS | | iTransformer | | PatchTST | | TSMixer | | Crossformer | | TimesNet | | DLinear | | FEDformer | |
|---|---|---|---|---|---|---|---|---|---|---|---|---|---|---|---|---|---|---|
| Metric | MSE | MAE | MSE | MAE | MSE | MAE | MSE | MAE | MSE | MAE | MSE | MAE | MSE | MAE | MSE | MAE | MSE | MAE |
| ETTm1 | **0.393** | **0.402** | 0.397 | 0.405 | 0.407 | 0.410 | 0.396 | 0.406 | 0.398 | 0.407 | 0.513 | 0.496 | 0.400 | 0.406 | 0.403 | 0.407 | 0.448 | 0.452 |
| ETTm2 | **0.282** | **0.326** | 0.287 | 0.330 | 0.288 | 0.332 | 0.287 | 0.330 | 0.289 | 0.333 | 0.757 | 0.610 | 0.291 | 0.333 | 0.350 | 0.401 | 0.305 | 0.349 |
| ETTh1 | 0.451 | **0.442** | 0.453 | 0.446 | 0.454 | 0.447 | 0.453 | 0.446 | 0.463 | 0.452 | 0.529 | 0.522 | 0.458 | 0.450 | 0.456 | 0.452 | **0.440** | 0.460 |
| ETTh2 | **0.380** | **0.405** | 0.384 | 0.406 | 0.383 | 0.407 | 0.385 | 0.410 | 0.401 | 0.417 | 0.942 | 0.684 | 0.414 | 0.427 | 0.559 | 0.515 | 0.437 | 0.449 |

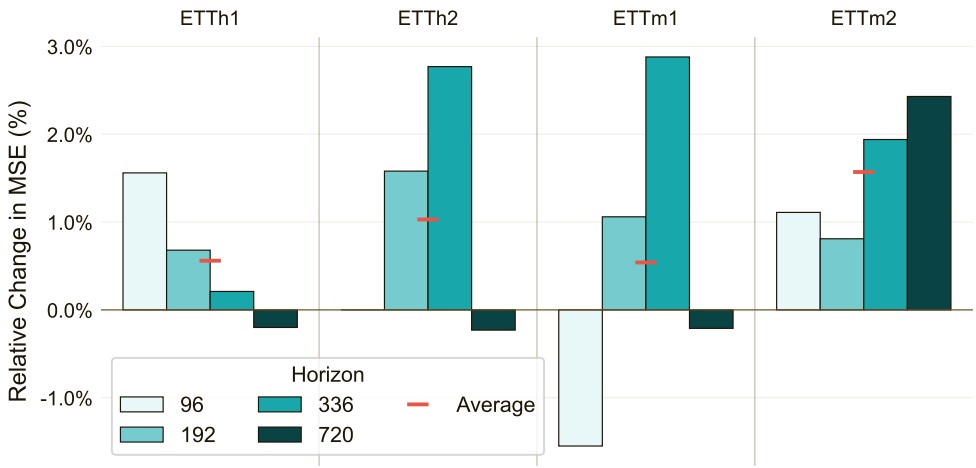

Figure 6: MSE performance of I-SOFTS over SOFTS across. Bars show the relative change in MSE. Positive values favor I-SOFTS.

#### 4.2.1 Performance of I-SOFTS on Larger Datasets

I-SOFTS failed to maintain performance on larger benchmarks (Table 4). I-SOFTS underperforms the SOFTS baseline on every dataset, with degradation particularly severe on the PEMS family (28.2—55.8%). Weather had the least degradation of -0.44%. These results indicate that some form of channel interaction is essential for some datasets. While the optimal degree of mixing may vary with dataset characteristics, eliminating it is never beneficial when the data contains exploitable spatial correlations.

Table 4: Relative performance of I-SOFTS on larger channel datasets

| Dataset | Weather | ECL | Traffic | Solar | PEMS03 | PEMS04 | PEMS07 | PEMS08 |
|---|---|---|---|---|---|---|---|---|
| Avg MSE Δ% | -0.44 | -7.17 | -9.64 | -9.47 | -28.18 | -49.52 | -55.76 | -31.92 |

## 5 Discussion

The central thesis of this work is that model selection should be driven by dataset characteristics, not architectural complexity. Using SOFTS, we demonstrate that extreme models, i.e., fully CI or CD, often outperform their hybrid counterparts when their inductive bias aligns with the dataset's spatial correlation structure.

### 5.1 Spatial Structure Determines Model Effectiveness

I-SOFTS achieves competitive performance across all ETT datasets, consistently outperforming SOFTS with stable gains (Figure 6).However, it degrades on larger datasets (Table 4).

C-SOFTS performs strongest in homogeneous regimes with moderate cluster organization. PEMS03, PEMS04, and PEMS08 achieve MSE improvements of 7.3%, 15.9%, and 5.2%, respectively, while block-structured ECL and Weather improve by 4.3% and 3.0%. On PEMS07, the full C-SOFTS model degrades by -1.71% relative to SOFTS, but ablating the spatial mixer improves MSE by 12.7%. This confirms that the spatial mixer is the active liability on datasets with many clusters, and that this study's regime classification has mechanistic validity. On Traffic, the spatial mixer forces 862 incoherently organized channels through a shared projection, actively destroying structure and producing a -19.5% degradation. On Solar, channels are near-perfectly uniform, and the model finds no exploitable variation beyond what SOFTS already captures, producing a near-neutral -0.32%.

In summary, I-SOFTS offers competitive performance within the ETT dataset family, suggesting that the STAR module's channel mixing provides limited benefit in this low-channel, low-correlation regime. C-SOFTS dominates when the spatial structure is either homogeneous with coherent clustering or exhibits strong block separation. SOFTS is preferred only when the spatial structure is weak or incoherent.

### 5.2 Importance of the Dataset Descriptors

Prior studies have shown that CD models benefit from datasets with strong correlations, yet this claim has rarely been quantified. The high-correlation fraction provides such a measure. However, within the three variants used in this study, correlation magnitude alone is insufficient to predict C-SOFTS performance, and the structural organization of that correlation is equally consequential. This is best illustrated by contrasting Solar and Weather. Solar has the highest high-correlation fraction of 1, yet C-SOFTS marginally degrades, whilst Weather has the lowest high-correlation fraction of 0.25, yet C-SOFTS consistently improves across all horizons. Block separation, which measures how correlations are organized, explains this divergence. Weather has the highest block separation of 0.52, which means that its channels partition into groups with

meaningful internal contrast. Solar's lowest block separation of 0.05 indicates that its correlation is globally uniform, and the additional channel mixing provides little benefit over the hybrid approach.

The PEMS and Traffic datasets further refine these findings. Despite low block separation scores, the PEMS datasets achieve significant MSE improvements. What they do exhibit, however, are extremely high correlation fractions ($\geq$ 0.95). PEMS03 has a fraction score of 1, similar to Solar, but a higher block separation score (0.18 compared to Solar's 0.05). This suggests that when correlations are extremely high, even weakly structured spatial information is sufficient for the CD C-SOFTS model to exploit. Block separation, therefore, appears most consequential in moderate-correlation regimes, such as Traffic. Although Traffic's block separation score (0.18) is comparable to that of PEMS07, its lower fraction score (0.67 vs. 0.95) corresponds with performance degradation.

Beyond explaining performance differences, these descriptors have practical utility. They serve as a diagnostic guide for selecting the appropriate variant, and if that variant is C-SOFTS, for determining whether the spatial mixer should be included.

## 5.3  Implications for Model Development

The question "which architecture is best?" may be less useful than "which data characteristics favour which strategies?" Prior work frames the CI-CD tradeoff as a tension between robustness and capacity (Han et al., 2024b). Our results show that the optimal point on that spectrum is not a fixed architectural property but shifts with the dataset's spatial structure.

The findings of the previous section speak to a broader tension in the forecasting literature. Comprehensive benchmarks consistently show that no single architecture dominates across all datasets (Brigato et al., 2026; Li et al., 2025). Yet models are still routinely ranked solely by average performance across multiple benchmarks without assessing the conditions under which they perform. Our results, though limited to variants of a single architecture, suggest that this regime-dependence may be fundamental and is obscured by aggregate rankings. Within the SOFTS family, a variant that excels in homogeneous high-correlation regimes and degrades in incoherent ones is poorly described by its average rank; it is better described by the conditions under which it works. Whether this holds more broadly across architectures remains an open question, but it motivates a shift in how model comparisons are framed.

A more productive framing would expand the evaluation of models beyond whether they improve over baselines to include the spatial conditions under which those improvements occur. Reporting spatial properties alongside performance metrics, as we do in Table 2, is one concrete step in this direction. A fuller formalization would require a taxonomy of spatial regimes validated across a broader set of architectures and datasets, establishing not just that regime dependence exists but also which structural properties are architecturally diagnostic and which are incidental to the datasets studied here.

## 5.4  Limitations and Future Works

The proposed data descriptors are computed from the data independently of any model, but were validated exclusively within the SOFTS architecture and may not fully explain performance variations across broader model families. For example, on the Weather dataset, the CI PatchTST model achieves lower MSE than the CD iTransformer model (Table 1), despite Weather belonging to a regime which favoured C-SOFTS. Developing spatial descriptors that generalize across model families and evaluating them on a broader set of CI and CD architectures remains an important direction for future research.

While high-correlation fraction and block separation provide interpretable spatial diagnostics, this work does not attempt to derive formal decision boundaries between regimes. Establishing statistically robust thresholds would require a substantially larger and more diverse collection of datasets, probably synthetically generated, than the twelve benchmarks considered here. Future work should construct or curate datasets that systematically span the space of spatial correlation structures, enabling the derivation and validation of quantitative decision thresholds.

## 6 Conclusion

This work demonstrates that the effective performance of CI, CD, and hybrid models in time-series forecasting is not an inherent architectural property but depends on a dataset's spatial correlation structure. We demonstrate this by pushing the hybrid SOFTS model to its CI and CD extremes using I-SOFTS and C-SOFTS. Results show that simpler extreme configurations can outperform general-purpose hybrids when their inductive biases align with the underlying data regime.

We introduced two dataset properties, high-correlation fraction and block separation, to quantify dataset correlation and define boundaries for model preference. C-SOFTS is optimal on homogeneous or block-structured datasets with clear cluster contrast, while I-SOFTS is competitive on few-channel datasets. SOFTS is preferable when channels exhibit moderate correlation with low cluster organization. These findings suggest that pursuing a universal, one-size-fits-all forecasting architecture may be misguided. Instead, we advocate for regime-aware model development, where models are designed and evaluated for datasets with specific properties.

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

## A Datasets

We evaluated the models on twelve benchmark datasets (Table 5).

## B Implementation Details

### B.1 Dataset Descriptors

Algorithm 1 details the computation of the high-correlation fraction. Algorithm 2 details the computation of block separation, including the hierarchical clustering procedure and silhouette-based cluster count selection.

---

**Algorithm 1** Computation of High-Correlation Fraction

---

**Require:** Data matrix $X \in \mathbb{R}^{S \times C}$ (samples × channels), threshold $\tau = 0.5$.
**Ensure:** High-correlation fraction, $HCF \in [0, 1]$.
 1: **Preprocessing:**
 2: Compute Pearson correlation matrix $r \in \mathbb{R}^{C \times C}$ from $X$
 3: Let $\mathcal{P} = \{(i, j) \mid 1 \leq i < j \leq C\}$ be the set of unique channel pairs
 4: $N \leftarrow |\mathcal{P}| = C(C-1)/2$         ▷ Total number of unique channel pairs
 5: **Calculation:**
 6: $HCF \leftarrow \frac{1}{N} \sum_{(i,j) \in \mathcal{P}} \mathbb{I}(|r_{ij}| > \tau)$         ▷ $\mathbb{I}$ is the indicator function
 7: **return** $HCF$

---

Table 5: Dataset summary. Domain specifies the application area. Channels indicate the number of variables. Temporal resolution presents the sampling interval. Horizon specifies the future time steps predicted.

| Domain | Dataset | Channels | Temporal Resolution | Horizon |
|---|---|---|---|---|
| Energy | ETTh1, ETTh2 | 7 | 1 hour | 96, 192, 336, 720 |
| | ETTm1, ETTm2 | 7 | 15 minutes | 96, 192, 336, 720 |
| | ECL | 321 | 1 hour | 96, 192, 336, 720 |
| | Solar | 137 | 10 minutes | 96, 192, 336, 720 |
| Climate | Weather | 21 | 10 minutes | 96, 192, 336, 720 |
| Transport | Traffic | 862 | 1 hour | 96, 192, 336, 720 |
| | PEMS03 | 358 | 5 minutes | 12, 24, 48, 96 |
| | PEMS04 | 307 | 5 minutes | 12, 24, 48, 96 |
| | PEMS07 | 883 | 5 minutes | 12, 24, 48, 96 |
| | PEMS08 | 170 | 5 minutes | 12, 24, 48, 96 |

---

**Algorithm 2** Computation of Block Separation

---

**Require:** Data matrix $X \in \mathbb{R}^{S \times C}$, search range $K \leftarrow [2, \ \min(\lfloor C/2 \rfloor, 500) - 1]$.
**Ensure:** Block separation $\Delta_{wb}$, optimal cluster count $k^{\star}$.
1: **Preprocessing & Hierarchical Linkage:**
2: Compute Pearson correlation matrix $r \in \mathbb{R}^{C \times C}$ from $X$
3: Let $\mathcal{P} = \{(i,j) \mid 1 \leq i < j \leq C\}$          $\triangleright$ Set of unique channel pairs
4: $D_{ij} \leftarrow 1 - |r_{ij}| \quad \forall i,j \in [1, C]$          $\triangleright$ Define correlation distance
5: $D \leftarrow \frac{1}{2}(D + D^{\top})$         $\triangleright$ Symmetrize for numerical stability
6: $D \leftarrow \text{Clip}(D, 0, 2)$            $\triangleright$ Ensure valid bounds
7: $Z \leftarrow \text{WardLinkage}(D)$          $\triangleright$ Compute dendrogram once
8: **Optimize Cluster Count:**
9: best_score $\leftarrow -\infty$, $k^{\star} \leftarrow 2$
10: **for** $k$ **in** $K$ **do**
11:   labels$_k \leftarrow \text{FlatCluster}(Z, k)$     $\triangleright$ Extract $k$ clusters from precomputed dendrogram
12:   score $\leftarrow \text{SilhouetteScore}(D, \text{labels}_k, \text{metric} = \text{'precomputed'})$
13:   **if** score $>$ best_score **then**
14:    best_score $\leftarrow$ score
15:    $k^{\star} \leftarrow k$
16:   **end if**
17: **end for**
18: **Evaluate Block Cohesion and Separation:**
19: final_labels $\leftarrow \text{FlatCluster}(Z, k^{\star})$
20: $\mathcal{W} \leftarrow \{|r_{ij}| \mid (i,j) \in \mathcal{P} \text{ and final\_labels}_i = \text{final\_labels}_j\}$   $\triangleright$ Within-cluster absolute corr
21: $\mathcal{B} \leftarrow \{|r_{ij}| \mid (i,j) \in \mathcal{P} \text{ and final\_labels}_i \neq \text{final\_labels}_j\}$   $\triangleright$ Between-cluster absolute corr
22: $\mu_{within} \leftarrow \text{Mean}(\mathcal{W})$
23: $\mu_{between} \leftarrow \text{Mean}(\mathcal{B})$
24: $\Delta_{wb} \leftarrow \mu_{within} - \mu_{between}$          $\triangleright$ Block separation
25: **return** $\Delta_{wb}, k^{\star}$

---

### B.1.1 Stability of Data Descriptors Under Temporal Split

To assess whether these descriptors remain stable over time, we recomputed both metrics on the first 60% and last 40% of each dataset separately and report the differences in Table 6. The differences are small across all datasets (max $|\Delta\text{HCF}| = 0.14$, max $|\Delta\text{BS}| = 0.08$), and no dataset crosses a regime boundary as a result of these shifts. These suggest show that the high-correlation fraction and block separation descriptors reflect stable structural properties of the spatial correlation.

Table 6: Difference in high-correlation fraction and block separation between the first 60% and last 40% temporal splits of each dataset. Small differences suggests that both descriptors are stable across time and are not sensitive to temporal non-stationarity or concept drift.

| Dataset | $|\Delta\text{HCF}|$ | $|\Delta\text{BS}|$ | Dataset | $|\Delta\text{HCF}|$ | $|\Delta\text{BS}|$ |
|---------|------|------|---------|------|------|
| ETTh1 | 0.04 | 0.07 | PEMS03 | 0.00 | 0.02 |
| ETTh2 | -0.14 | -0.08 | PEMS04 | 0.01 | -0.03 |
| ETTm1 | 0.04 | 0.07 | PEMS07 | 0.01 | 0.00 |
| ETTm2 | -0.14 | -0.08 | PEMS08 | 0.02 | -0.06 |
| ECL | -0.01 | 0.05 | Traffic | 0.04 | -0.04 |
| Solar | 0.00 | 0.00 | Weather | -0.02 | 0.01 |

### B.2 Channel Mixer Architecture

Algorithm 3 details the forward pass of the Channel Mixer, covering the spatial mixing step, frequency-domain transformation, learned spectral filtering, and reconstruction. We use an orthonormal RFFT, which preserves energy; because all frequency modes are retained ($K_{\text{used}} = K$), the full transformation is energy-preserving.

## C   Full Results

### C.1 Full Results of Forecasting on Benchmark Datasets

The complete performance results of C-SOFTS and I-SOFTS compared to SOFTS and other forecasting models are presented in Tables 7 and 8, respectively. All forecasts use a lookback window of $L = 96$. The results for the hybrid SOFTS model were reproduced in this study, while the results for the other models were taken from the SOFTS paper Han et al. (2024a).

### C.2 Relative Performance

The relative performances of C-SOFTS and I-SOFTS against the baseline SOFTS hybrid models are presented in Tables 9 and 10, showing percentage changes in MSE across different datasets and forecasting horizons. Positive values indicate that the SOFTS variants achieve lower MSE than SOFTS, while negative values indicate worse performance.

### C.3 Ablation

The results of the spatial mixer ablation are presented in Table 11, showing the influence of each component, particularly the spatial mixer, on forecasting performance across different datasets.

Table 7: Results of C-SOFTS on eight benchmark datasets with a lookback window $L = 96$ and varying prediction lengths. The best results are bold and in red, and the second-best results are underlined and in blue.

| Models | | C-SOFTS | | SOFTS | | iTransformer | | PatchTST | | TSMixer | | Crossformer | | DLinear | | SCINet | | FEDformer | |
|---|---|---|---|---|---|---|---|---|---|---|---|---|---|---|---|---|---|---|---|
| Metric | | MSE | MAE | MSE | MAE | MSE | MAE | MSE | MAE | MSE | MAE | MSE | MAE | MSE | MAE | MSE | MAE | MSE | MAE |
| ECL | 96 | 0.142 | 0.241 | 0.145 | 0.236 | 0.148 | 0.240 | 0.164 | 0.251 | 0.157 | 0.260 | 0.219 | 0.314 | 0.197 | 0.282 | 0.247 | 0.245 | 0.193 | 0.308 |
| | 192 | 0.160 | 0.259 | 0.161 | 0.251 | 0.162 | 0.253 | 0.173 | 0.262 | 0.173 | 0.262 | 0.231 | 0.322 | 0.196 | 0.285 | 0.257 | 0.355 | 0.201 | 0.315 |
| | 336 | 0.175 | 0.276 | 0.181 | 0.272 | 0.178 | 0.269 | 0.190 | 0.279 | 0.190 | 0.279 | 0.246 | 0.337 | 0.209 | 0.301 | 0.269 | 0.369 | 0.214 | 0.329 |
| | 720 | 0.193 | 0.295 | 0.217 | 0.305 | 0.225 | 0.317 | 0.230 | 0.313 | 0.230 | 0.313 | 0.280 | 0.363 | 0.245 | 0.333 | 0.299 | 0.390 | 0.246 | 0.355 |
| | Avg | 0.168 | 0.268 | 0.176 | 0.266 | 0.178 | 0.270 | 0.189 | 0.276 | 0.189 | 0.276 | 0.244 | 0.334 | 0.212 | 0.300 | 0.268 | 0.365 | 0.214 | 0.327 |
| Traffic | 96 | 0.456 | 0.289 | 0.375 | 0.254 | 0.395 | 0.268 | 0.427 | 0.272 | 0.493 | 0.336 | 0.522 | 0.290 | 0.650 | 0.396 | 0.788 | 0.499 | 0.587 | 0.366 |
| | 192 | 0.467 | 0.314 | 0.398 | 0.261 | 0.417 | 0.276 | 0.454 | 0.289 | 0.497 | 0.351 | 0.530 | 0.293 | 0.598 | 0.370 | 0.789 | 0.505 | 0.604 | 0.373 |
| | 336 | 0.494 | 0.329 | 0.416 | 0.269 | 0.433 | 0.283 | 0.450 | 0.282 | 0.528 | 0.361 | 0.558 | 0.305 | 0.605 | 0.373 | 0.797 | 0.508 | 0.621 | 0.383 |
| | 720 | 0.540 | 0.349 | 0.449 | 0.288 | 0.467 | 0.302 | 0.484 | 0.301 | 0.569 | 0.380 | 0.589 | 0.328 | 0.645 | 0.394 | 0.841 | 0.523 | 0.626 | 0.382 |
| | Avg | 0.489 | 0.320 | 0.410 | 0.268 | 0.428 | 0.282 | 0.454 | 0.286 | 0.522 | 0.357 | 0.550 | 0.304 | 0.625 | 0.383 | 0.804 | 0.509 | 0.610 | 0.376 |
| Weather | 96 | 0.162 | 0.208 | 0.171 | 0.213 | 0.174 | 0.214 | 0.176 | 0.217 | 0.166 | 0.210 | 0.158 | 0.230 | 0.196 | 0.255 | 0.221 | 0.306 | 0.217 | 0.296 |
| | 192 | 0.215 | 0.257 | 0.221 | 0.256 | 0.221 | 0.254 | 0.221 | 0.256 | 0.215 | 0.256 | 0.206 | 0.277 | 0.237 | 0.296 | 0.261 | 0.340 | 0.276 | 0.336 |
| | 336 | 0.271 | 0.295 | 0.278 | 0.298 | 0.278 | 0.296 | 0.275 | 0.296 | 0.287 | 0.300 | 0.272 | 0.335 | 0.283 | 0.335 | 0.309 | 0.378 | 0.339 | 0.380 |
| | 720 | 0.351 | 0.344 | 0.356 | 0.349 | 0.358 | 0.347 | 0.352 | 0.346 | 0.355 | 0.348 | 0.398 | 0.418 | 0.345 | 0.381 | 0.377 | 0.427 | 0.403 | 0.428 |
| | Avg | 0.250 | 0.276 | 0.256 | 0.279 | 0.258 | 0.278 | 0.256 | 0.279 | 0.256 | 0.279 | 0.259 | 0.315 | 0.265 | 0.317 | 0.292 | 0.363 | 0.309 | 0.360 |
| Solar | 96 | 0.198 | 0.239 | 0.202 | 0.230 | 0.203 | 0.237 | 0.205 | 0.246 | 0.221 | 0.275 | 0.310 | 0.331 | 0.290 | 0.378 | 0.237 | 0.344 | 0.242 | 0.342 |
| | 192 | 0.227 | 0.262 | 0.230 | 0.254 | 0.233 | 0.261 | 0.237 | 0.267 | 0.268 | 0.306 | 0.734 | 0.725 | 0.320 | 0.398 | 0.280 | 0.380 | 0.285 | 0.380 |
| | 336 | 0.254 | 0.284 | 0.241 | 0.268 | 0.248 | 0.273 | 0.250 | 0.276 | 0.272 | 0.294 | 0.750 | 0.735 | 0.353 | 0.415 | 0.304 | 0.389 | 0.282 | 0.376 |
| | 720 | 0.243 | 0.275 | 0.245 | 0.273 | 0.249 | 0.275 | 0.252 | 0.275 | 0.281 | 0.313 | 0.769 | 0.765 | 0.356 | 0.413 | 0.308 | 0.388 | 0.357 | 0.427 |
| | Avg | 0.231 | 0.265 | 0.230 | 0.256 | 0.233 | 0.262 | 0.236 | 0.266 | 0.260 | 0.297 | 0.641 | 0.639 | 0.330 | 0.401 | 0.282 | 0.375 | 0.291 | 0.381 |
| PEMS03 | 12 | 0.068 | 0.167 | 0.068 | 0.171 | 0.071 | 0.174 | 0.073 | 0.178 | 0.075 | 0.186 | 0.090 | 0.203 | 0.122 | 0.243 | 0.066 | 0.172 | 0.126 | 0.251 |
| | 24 | 0.081 | 0.179 | 0.085 | 0.189 | 0.093 | 0.201 | 0.105 | 0.212 | 0.095 | 0.210 | 0.121 | 0.240 | 0.201 | 0.317 | 0.085 | 0.198 | 0.149 | 0.275 |
| | 48 | 0.104 | 0.205 | 0.116 | 0.225 | 0.125 | 0.236 | 0.159 | 0.264 | 0.121 | 0.240 | 0.202 | 0.317 | 0.333 | 0.425 | 0.127 | 0.238 | 0.227 | 0.348 |
| | 96 | 0.133 | 0.236 | 0.157 | 0.263 | 0.164 | 0.275 | 0.210 | 0.305 | 0.184 | 0.295 | 0.262 | 0.367 | 0.457 | 0.515 | 0.178 | 0.287 | 0.348 | 0.434 |
| | Avg | 0.097 | 0.197 | 0.107 | 0.212 | 0.113 | 0.221 | 0.137 | 0.240 | 0.119 | 0.233 | 0.169 | 0.281 | 0.278 | 0.375 | 0.114 | 0.224 | 0.213 | 0.327 |
| PEMS04 | 12 | 0.067 | 0.165 | 0.074 | 0.175 | 0.078 | 0.183 | 0.085 | 0.189 | 0.079 | 0.188 | 0.098 | 0.218 | 0.148 | 0.272 | 0.073 | 0.177 | 0.138 | 0.262 |
| | 24 | 0.076 | 0.176 | 0.088 | 0.193 | 0.095 | 0.205 | 0.115 | 0.222 | 0.089 | 0.201 | 0.131 | 0.256 | 0.224 | 0.340 | 0.084 | 0.193 | 0.177 | 0.293 |
| | 48 | 0.090 | 0.198 | 0.110 | 0.218 | 0.120 | 0.233 | 0.167 | 0.273 | 0.111 | 0.222 | 0.205 | 0.326 | 0.355 | 0.437 | 0.099 | 0.211 | 0.270 | 0.368 |
| | 96 | 0.107 | 0.217 | 0.138 | 0.245 | 0.150 | 0.262 | 0.211 | 0.310 | 0.133 | 0.247 | 0.402 | 0.457 | 0.452 | 0.504 | 0.114 | 0.227 | 0.341 | 0.427 |
| | Avg | 0.085 | 0.189 | 0.103 | 0.208 | 0.111 | 0.221 | 0.145 | 0.249 | 0.103 | 0.215 | 0.209 | 0.314 | 0.295 | 0.388 | 0.092 | 0.202 | 0.231 | 0.337 |
| PEMS07 | 12 | 0.064 | 0.146 | 0.058 | 0.151 | 0.067 | 0.165 | 0.068 | 0.163 | 0.073 | 0.181 | 0.094 | 0.200 | 0.115 | 0.242 | 0.068 | 0.171 | 0.109 | 0.225 |
| | 24 | 0.078 | 0.158 | 0.075 | 0.172 | 0.088 | 0.190 | 0.102 | 0.201 | 0.090 | 0.199 | 0.139 | 0.247 | 0.210 | 0.329 | 0.119 | 0.225 | 0.125 | 0.244 |
| | 48 | 0.098 | 0.182 | 0.098 | 0.197 | 0.110 | 0.215 | 0.170 | 0.261 | 0.124 | 0.231 | 0.311 | 0.369 | 0.398 | 0.458 | 0.149 | 0.237 | 0.165 | 0.288 |
| | 96 | 0.111 | 0.192 | 0.120 | 0.216 | 0.139 | 0.245 | 0.236 | 0.308 | 0.163 | 0.255 | 0.396 | 0.442 | 0.594 | 0.553 | 0.141 | 0.234 | 0.262 | 0.376 |
| | Avg | 0.088 | 0.170 | 0.088 | 0.184 | 0.101 | 0.204 | 0.144 | 0.233 | 0.112 | 0.217 | 0.235 | 0.315 | 0.329 | 0.395 | 0.119 | 0.234 | 0.165 | 0.283 |
| PEMS08 | 12 | 0.072 | 0.172 | 0.075 | 0.172 | 0.079 | 0.182 | 0.098 | 0.205 | 0.083 | 0.189 | 0.165 | 0.214 | 0.154 | 0.276 | 0.087 | 0.184 | 0.173 | 0.273 |
| | 24 | 0.097 | 0.195 | 0.105 | 0.202 | 0.115 | 0.219 | 0.162 | 0.266 | 0.117 | 0.226 | 0.215 | 0.260 | 0.248 | 0.353 | 0.122 | 0.221 | 0.210 | 0.301 |
| | 48 | 0.143 | 0.226 | 0.163 | 0.251 | 0.186 | 0.235 | 0.238 | 0.311 | 0.196 | 0.299 | 0.315 | 0.355 | 0.440 | 0.470 | 0.189 | 0.270 | 0.320 | 0.394 |
| | 96 | 0.223 | 0.248 | 0.216 | 0.257 | 0.221 | 0.267 | 0.303 | 0.318 | 0.266 | 0.331 | 0.377 | 0.397 | 0.674 | 0.565 | 0.236 | 0.300 | 0.442 | 0.465 |
| | Avg | 0.134 | 0.210 | 0.140 | 0.220 | 0.150 | 0.226 | 0.200 | 0.275 | 0.165 | 0.261 | 0.268 | 0.307 | 0.379 | 0.416 | 0.158 | 0.244 | 0.286 | 0.358 |
| 1st Count | | 26 | 25 | 12 | 13 | 0 | 2 | 0 | 0 | 0 | 0 | 2 | 0 | 1 | 0 | 1 | 0 | 0 | 0 |

---

**Algorithm 3** Channel Mixer Forward Pass

---

**Require:** $x \in \mathbb{R}^{B \times C \times d}$, spatial mixer weights, scale, mode_gate
**Ensure:** $x_{\text{out}} \in \mathbb{R}^{B \times C \times d}$

 1: **Spatial Mixing:**
 2: $x \leftarrow x^{\top}$          $\triangleright (B, C, d) \rightarrow (B, d, C)$
 3: $x \leftarrow \text{Linear}(C \rightarrow r)(x)$
 4: $x \leftarrow \text{ReLU}(x)$
 5: $x \leftarrow \text{Linear}(r \rightarrow C)(x)$
 6: $x \leftarrow x^{\top}$          $\triangleright (B, d, C) \rightarrow (B, C, d)$
 7: **Frequency-Domain Transformation:**
 8: $x_{\text{fft}} \leftarrow \text{RFFT}(x, \dim = 1)$          $\triangleright (B, K, d),\; K = \text{floor}(C/2) + 1$
 9: $K_{\text{used}} \leftarrow \min(K, x_{\text{fft}}.\text{shape}[1])$        $\triangleright = K$ for standard input; all modes retained
10: $W \leftarrow \text{view\_as\_complex}(\text{scale})[: K_{\text{used}}, :]$
11: $G \leftarrow \text{sigmoid}(\text{mode\_gate})[: K_{\text{used}}, :, :]$          $\triangleright \text{mode\_gate} \in \mathbb{R}^{K \times 1 \times 1}$
12: **Frequency Filtering:**
13: $x_{\text{fft\_out}} \leftarrow 0$          $\triangleright$ Initialize zeros like $x_{\text{fft}}$
14: $x_{\text{fft\_out}}[:, : K_{\text{used}}, :] \leftarrow x_{\text{fft}}[:, : K_{\text{used}}, :] \odot W \odot G$
15: **Inverse Frequency-Domain Transformation:**
16: $x_{\text{out}} \leftarrow \text{IRFFT}(x_{\text{fft\_out}}, n = C, \dim = 1)$
17: $x_{\text{out}} \leftarrow \text{Dropout}(x_{\text{out}})$
18: **return** $x_{\text{out}}$

---

Table 8: Results of I-SOFTS on four ETT datasets. The best results are bold and in red, and the second-best results are underlined and in blue.

| Models | | I-SOFTS | | SOFTS | | iTransformer | | PatchTST | | TSMixer | | Crossformer | | TimesNet | | DLinear | | FEDformer | |
|---|---|---|---|---|---|---|---|---|---|---|---|---|---|---|---|---|---|---|---|---|
| Metric | | MSE | MAE | MSE | MAE | MSE | MAE | MSE | MAE | MSE | MAE | MSE | MAE | MSE | MAE | MSE | MAE | MSE | MAE |
| ETTm1 | 96 | 0.327 | 0.365 | **0.322** | **0.361** | 0.334 | 0.368 | 0.329 | 0.365 | 0.323 | 0.363 | 0.404 | 0.426 | 0.338 | 0.375 | 0.345 | 0.372 | 0.379 | 0.419 |
| | 192 | **0.374** | 0.388 | 0.378 | 0.392 | 0.377 | 0.391 | 0.380 | 0.394 | 0.376 | 0.392 | 0.450 | 0.451 | **0.374** | **0.387** | 0.380 | 0.389 | 0.426 | 0.441 |
| | 336 | 0.404 | **0.409** | 0.416 | 0.417 | 0.426 | 0.420 | **0.400** | 0.410 | 0.407 | 0.413 | 0.532 | 0.515 | 0.410 | 0.411 | 0.413 | 0.413 | 0.445 | 0.459 |
| | 720 | 0.469 | **0.449** | **0.468** | **0.449** | 0.491 | 0.459 | 0.475 | 0.453 | 0.485 | 0.459 | 0.666 | 0.589 | 0.478 | 0.450 | 0.474 | 0.453 | 0.543 | 0.490 |
| | Avg | **0.393** | **0.402** | 0.397 | 0.405 | 0.407 | 0.410 | 0.396 | 0.406 | 0.398 | 0.407 | 0.513 | 0.496 | 0.400 | 0.406 | 0.403 | 0.407 | 0.448 | 0.452 |
| ETTm2 | 96 | **0.178** | **0.260** | 0.180 | 0.262 | 0.180 | 0.264 | 0.184 | 0.264 | 0.182 | 0.266 | 0.287 | 0.366 | 0.187 | 0.267 | 0.193 | 0.292 | 0.203 | 0.287 |
| | 192 | **0.244** | **0.304** | 0.246 | 0.306 | 0.250 | 0.309 | 0.246 | 0.306 | 0.249 | 0.309 | 0.414 | 0.492 | 0.249 | 0.309 | 0.284 | 0.362 | 0.269 | 0.328 |
| | 336 | **0.303** | **0.340** | 0.309 | 0.346 | 0.311 | 0.348 | 0.308 | 0.346 | 0.309 | 0.347 | 0.597 | 0.542 | 0.321 | 0.351 | 0.369 | 0.427 | 0.325 | 0.366 |
| | 720 | **0.401** | **0.399** | 0.411 | 0.405 | 0.412 | 0.407 | 0.409 | 0.402 | 0.416 | 0.408 | 1.730 | 1.042 | 0.408 | 0.403 | 0.554 | 0.522 | 0.421 | 0.415 |
| | Avg | **0.282** | **0.326** | 0.287 | 0.330 | 0.288 | 0.332 | 0.287 | 0.330 | 0.289 | 0.333 | 0.757 | 0.610 | 0.291 | 0.333 | 0.350 | 0.401 | 0.305 | 0.349 |
| ETTh1 | 96 | 0.378 | **0.397** | 0.384 | 0.403 | 0.386 | 0.405 | 0.394 | 0.406 | 0.401 | 0.412 | 0.423 | 0.448 | 0.384 | 0.402 | 0.386 | 0.400 | **0.376** | 0.419 |
| | 192 | 0.435 | **0.428** | 0.438 | 0.432 | 0.441 | 0.436 | 0.440 | 0.435 | 0.452 | 0.442 | 0.471 | 0.474 | 0.436 | 0.429 | 0.437 | 0.432 | **0.420** | 0.448 |
| | 336 | 0.482 | **0.454** | 0.483 | 0.457 | 0.487 | 0.458 | 0.491 | 0.462 | 0.492 | 0.463 | 0.570 | 0.546 | 0.491 | 0.469 | 0.481 | 0.459 | **0.459** | 0.465 |
| | 720 | 0.508 | 0.491 | 0.507 | 0.493 | 0.503 | 0.491 | **0.487** | **0.479** | 0.507 | 0.490 | 0.653 | 0.621 | 0.521 | 0.500 | 0.519 | 0.516 | 0.506 | 0.507 |
| | Avg | 0.451 | **0.442** | 0.453 | 0.446 | 0.454 | 0.447 | 0.453 | 0.446 | 0.463 | 0.452 | 0.529 | 0.522 | 0.458 | 0.450 | 0.456 | 0.452 | **0.440** | 0.460 |
| ETTh2 | 96 | 0.297 | 0.346 | 0.297 | 0.348 | 0.297 | 0.349 | **0.288** | **0.340** | 0.319 | 0.361 | 0.745 | 0.584 | 0.340 | 0.374 | 0.333 | 0.387 | 0.358 | 0.397 |
| | 192 | **0.374** | 0.396 | 0.380 | 0.398 | 0.380 | 0.400 | 0.376 | **0.395** | 0.402 | 0.410 | 0.877 | 0.656 | 0.402 | 0.414 | 0.477 | 0.476 | 0.429 | 0.439 |
| | 336 | **0.421** | **0.432** | 0.433 | 0.437 | 0.428 | **0.432** | 0.440 | 0.451 | 0.444 | 0.446 | 1.043 | 0.731 | 0.452 | 0.452 | 0.594 | 0.541 | 0.496 | 0.487 |
| | 720 | 0.427 | 0.445 | **0.426** | **0.442** | 0.427 | 0.445 | 0.436 | 0.453 | 0.441 | 0.450 | 1.104 | 0.763 | 0.462 | 0.468 | 0.831 | 0.657 | 0.463 | 0.474 |
| | Avg | **0.380** | **0.405** | 0.384 | 0.406 | 0.383 | 0.407 | 0.385 | 0.410 | 0.401 | 0.417 | 0.942 | 0.684 | 0.414 | 0.427 | 0.559 | 0.515 | 0.437 | 0.449 |
| 1st Count | | 10 | 14 | 3 | 3 | 0 | 1 | 3 | 3 | 0 | 0 | 0 | 0 | 1 | 1 | 0 | 0 | 4 | 0 |

Table 9: Relative performance of C-SOFTS compared to the baseline SOFTS model across datasets and forecasting horizons, measured as percentage change in MSE.

| Dataset | ECL | | Solar | | Weather | | Traffic | |
|---|---|---|---|---|---|---|---|---|
| Horizon | MSE | MAE | MSE | MAE | MSE | MAE | MSE | MAE |
| 96 | 2.1 | -2.1 | 2.0 | -3.9 | 5.3 | 2.4 | -21.6 | -13.8 |
| 192 | 0.6 | -3.2 | 1.3 | -3.2 | 2.7 | -0.4 | -17.3 | -20.3 |
| 336 | 3.3 | -1.5 | -5.4 | -6.0 | 2.5 | 1.0 | -18.8 | -22.3 |
| 720 | 11.1 | 3.3 | 0.8 | -0.7 | 1.4 | 1.4 | -20.3 | -21.2 |
| Avg | 4.3 | -0.9 | -0.3 | -3.4 | 3.0 | 1.1 | -19.5 | -19.4 |

| Dataset | PEMS03 | | PEMS04 | | PEMS07 | | PEMS08 | |
|---|---|---|---|---|---|---|---|---|
| Horizon | MSE | MAE | MSE | MAE | MSE | MAE | MSE | MAE |
| 12 | -1.5 | 1.8 | 9.5 | 5.7 | -10.3 | 3.3 | 4.0 | 0.0 |
| 24 | 4.7 | 5.3 | 13.6 | 8.8 | -4.0 | 8.1 | 9.4 | 4.9 |
| 48 | 11.1 | 9.3 | 18.2 | 9.2 | 0.0 | 7.6 | 12.8 | 10.7 |
| 96 | 14.7 | 9.9 | 22.5 | 11.4 | 7.5 | 11.1 | -3.2 | 3.9 |
| Avg | 7.3 | 6.6 | 15.9 | 8.8 | -1.7 | 7.5 | 5.7 | 4.9 |

Table 10: Relative performance of I-SOFTS compared to the baseline SOFTS model across datasets and forecasting horizons, measured as percentage change in MSE.

| Dataset | ETTm1 | | ETTm2 | | ETTh1 | | ETTh2 | |
|---|---|---|---|---|---|---|---|---|
| Horizon | MSE | MAE | MSE | MAE | MSE | MAE | MSE | MAE |
| 96 | -1.55 | -1.11 | 1.11 | 0.76 | 1.56 | 1.49 | 0.00 | 0.57 |
| 192 | 1.06 | 1.02 | 0.81 | 0.65 | 0.68 | 0.93 | 1.58 | 0.50 |
| 336 | 2.88 | 1.92 | 1.94 | 1.73 | 0.21 | 0.66 | 2.77 | 1.14 |
| 720 | -0.21 | 0.00 | 2.43 | 1.48 | -0.20 | 0.41 | -0.23 | -0.68 |
| Avg | 0.54 | 0.46 | 1.57 | 1.16 | 0.56 | 0.87 | 1.03 | 0.39 |

Table 11: Comparison of the effect of ablating core components of the Channel Mixer. The term "w/o" denotes "without" the corresponding component. For each row, the lowest MSE and MAE values are highlighted in bold red

| Ablation | | Full | | w/o Mode Gates | | w/o Scale | | w/o Spatial Mixer | |
|---|---|---|---|---|---|---|---|---|---|
| Metrics | | MSE | MAE | MSE | MAE | MSE | MAE | MSE | MAE |
| ECL | 96 | 0.146 | 0.244 | 0.144 | 0.244 | 0.149 | 0.250 | **0.136** | **0.233** |
| | 192 | 0.163 | 0.260 | 0.164 | 0.261 | 0.166 | 0.264 | **0.154** | **0.250** |
| | 336 | 0.173 | 0.276 | 0.177 | 0.278 | 0.184 | 0.280 | **0.170** | **0.267** |
| | 720 | 0.197 | 0.298 | 0.194 | 0.294 | 0.206 | 0.303 | **0.193** | **0.292** |
| | Avg | 0.170 | 0.270 | 0.170 | 0.269 | 0.176 | 0.274 | **0.163** | **0.261** |
| Traffic | 96 | 0.452 | 0.288 | 0.446 | 0.291 | 0.497 | 0.318 | **0.388** | **0.266** |
| | 192 | 0.470 | 0.316 | 0.472 | 0.318 | 0.494 | 0.338 | **0.409** | **0.274** |
| | 336 | 0.494 | 0.329 | 0.498 | 0.341 | 0.531 | 0.346 | **0.429** | **0.283** |
| | 720 | 0.541 | 0.350 | 0.546 | 0.355 | 0.595 | 0.368 | **0.485** | **0.305** |
| | Avg | 0.489 | 0.321 | 0.491 | 0.326 | 0.529 | 0.343 | **0.428** | **0.282** |
| Solar | 96 | 0.202 | 0.246 | 0.199 | 0.246 | 0.229 | 0.262 | **0.198** | **0.233** |
| | 192 | 0.228 | 0.267 | **0.226** | 0.267 | 0.258 | 0.272 | 0.229 | **0.259** |
| | 336 | 0.246 | 0.276 | 0.259 | 0.280 | 0.280 | 0.288 | **0.245** | **0.273** |
| | 720 | 0.244 | 0.277 | **0.243** | 0.277 | 0.256 | 0.282 | 0.248 | **0.276** |
| | Avg | **0.230** | 0.267 | 0.232 | 0.268 | 0.256 | 0.276 | **0.230** | **0.260** |
| PEMS04 | 12 | **0.067** | **0.164** | **0.067** | 0.166 | 0.075 | 0.180 | 0.068 | 0.170 |
| | 24 | 0.076 | 0.177 | **0.075** | **0.175** | 0.082 | 0.190 | 0.077 | 0.183 |
| | 48 | **0.089** | **0.194** | 0.091 | 0.196 | 0.095 | 0.206 | 0.091 | 0.201 |
| | 96 | **0.103** | **0.211** | 0.104 | 0.212 | 0.108 | 0.216 | 0.109 | 0.220 |
| | Avg | **0.084** | **0.187** | **0.084** | **0.187** | 0.090 | 0.198 | 0.086 | 0.194 |
| PEMS07 | 12 | 0.059 | **0.144** | 0.060 | 0.145 | 0.067 | 0.162 | **0.056** | 0.153 |
| | 24 | 0.083 | **0.159** | 0.080 | 0.162 | 0.085 | 0.182 | **0.070** | 0.171 |
| | 48 | 0.103 | 0.180 | 0.097 | **0.175** | 0.107 | 0.199 | **0.082** | **0.175** |
| | 96 | 0.116 | 0.201 | 0.123 | 0.198 | 0.132 | 0.220 | **0.105** | **0.194** |
| | Avg | 0.090 | 0.171 | 0.090 | **0.170** | 0.098 | 0.191 | **0.078** | 0.173 |
| PEMS08 | 12 | 0.072 | **0.171** | 0.072 | **0.171** | 0.075 | 0.180 | **0.071** | 0.172 |
| | 24 | **0.095** | **0.191** | **0.095** | 0.192 | 0.100 | 0.203 | 0.096 | 0.199 |
| | 48 | 0.153 | 0.247 | 0.140 | 0.223 | 0.150 | 0.239 | **0.127** | **0.221** |
| | 96 | **0.237** | 0.258 | 0.239 | **0.256** | 0.251 | 0.275 | 0.283 | 0.322 |
| | Avg | 0.139 | 0.217 | **0.137** | **0.211** | 0.144 | 0.224 | 0.144 | 0.229 |

## D  Sensitivity Analysis of the Spatial Mixer's Bottleneck Dimension

To evaluate the impact of spatial pre-processing before spectral filtering, we conducted a sensitivity analysis on the spatial mixer's hidden dimension. An initial broad sweep ($r \in \{16, 32, 64, C/2, C, 2C\}$) showed moderate sensitivity. From heavy compression ($r = 16$) to expansion ($r = 2C$), MSE only varied by 5.6% for PEMS04 and 6.9% for ECL. The finer-grained sequential sweep around representative values confirmed the non-monotonic behavior of the spatial mixer, with adjacent $r$ values producing up to 4.6% and 2.8% MSE variation for PEMS04 and ECL, respectively (Figure 9).

These findings highlight two key challenges in spatial mixer optimization. First, the optimal $r$ values are dataset-dependent and non-monotonic, making theoretical prediction difficult without dataset-specific empirical search. For example, ECL achieves the same MSE of 0.142 at $r = 98$ and $r = 642$. Second, empirical observations show that optimal $r$ values can vary across forecast horizons.

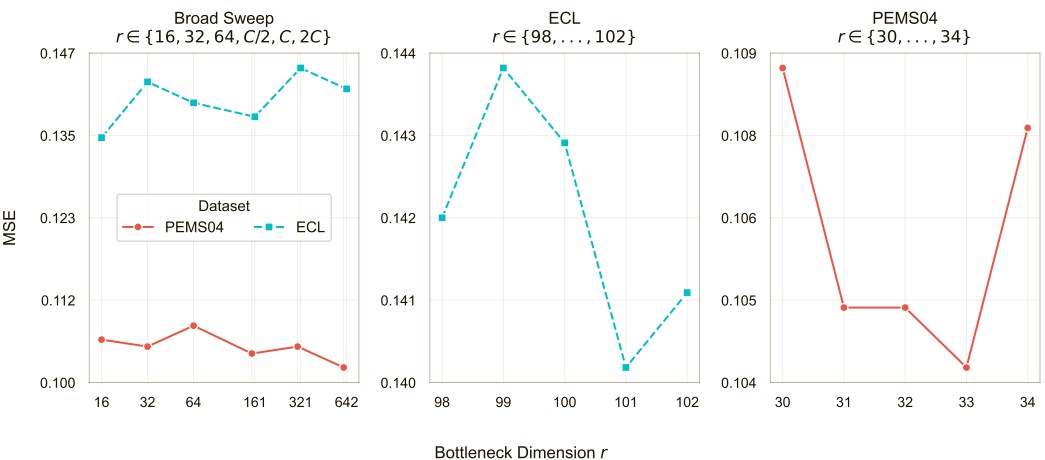

Figure 7: Sensitivity of MSE to spatial mixer's hidden dimension $r$ at $H = 96$. (*Left*) Broad sweep from heavy compression ($r = 16$) to expansion ($r = 2C$) for PEMS04 ($C = 307$) and ECL ($C = 321$). MSE varies by 5.6% and 6.9%, respectively. (*Center*) Fine-grained sequential sweep for ECL confirms non-monotonic behavior. (*Right*) Corresponding fine-grained sweep for PEMS04.

## E  Complexity Analysis

### E.1  Theoretical Complexity

The computational complexity of SOFTS scales as $O(CLd+Cd^2+CdH)$ with respect to the lookback window $L$, number of channels $C$, model dimension $d$, and forecasting horizon $H$. Reversible instance normalization incurs $O(CL)$, series embedding requires $O(CLd)$. Within each encoding layer, assuming the core dimension $d' = d$, the STAR module costs $O(Cd^2)$ for the two MLPs plus $O(Cd)$ for stochastic pooling. The FFN contributes another $O(Cd^2)$ per layer. The linear predictor adds $O(CdH)$. Overall, the architecture scales linearly with $C$, $L$, and $H$.

I-SOFTS retains the same asymptotic complexity as SOFTS: $O(CLd + Cd^2 + CdH)$. However, it eliminates the $O(Cd^2)$ cost of the STAR module. Its $O(Cd^2)$ comes from the Conv1D feedforward network. I-SOFTS achieves lower constant factors, reduces actual computational cost, and maintains linear scaling with respect to $C$, $L$, and $H$.

The overall complexity of C-SOFTS is $O(CLd + Cd^2 + Cdr + Cd\log C + CdH)$. The normalization and embedding stages, as well as the STAR module, remain unchanged, incurring $O(CL)$, $O(CLd)$, and $O(Cd^2)$,

respectively. The channel mixer module applies to a two-layer MLP across channels, mapping $C \to r \to C$. It yields a complexity of $O(Cdr)$, which simplifies asymptotically to $O(Cd)$ when $r$ is treated as a constant independent of $C$. The forward and inverse real FFT operations along the channel dimension each require $O(Cd\log C)$, resulting in $O(Cd\log C)$ overall up to constant factors. The spectral filtering stage performs element-wise complex multiplications over approximately $K \approx C/2$ modes, contributing $O(Kd) = O(Cd)$. Consequently, the dominant complexity introduced by Channel Mixer is $O(Cdr + Cd\log C)$.

Table 12 summarizes the key architectural differences and computational characteristics of SOFTS and its variants.

Table 12: Complexity Analysis of SOFTS and its variants.

|  | **SOFTS** | **C-SOFTS** | **I-SOFTS** |
| --- | --- | --- | --- |
| Channel Interaction | STAR | STAR and Channel Mixer | None |
| STAR Complexity | $O(Cd^2)$ | $O(Cd^2)$ | 0 |
| Post STAR Complexity | $O(Cd^2)$ | $O(Cdr + Cd\log C)$ | $O(Cd^2)$ |
| Total Complexity | $O(CLd+Cd^2+CdH)$ | $O(CLd+Cd^2+Cdr+ Cd\log C + CdH)$ | $O(CLd+Cd^2+CdH)$ |

## E.2 Empirical Benchmarks

Figure 8 presents the inference time, computational complexity, peak GPU memory, and parameter counts across channel counts.

I-SOFTS is the most efficient model. Removing the STAR module reduces parameter count by 31% relative to SOFTS (370K vs 535K). At $C = 5{,}000$, I-SOFTS's inference is twice as fast as that of SOFTS and C-SOFTS. C-SOFTS is comparable to SOFTS in GPU inference time while requiring fewer floating-point operations (FLOPs) at every scale. At $C = 5{,}000$, C-SOFTS requires 1.44B FLOPs compared to SOFTS's 2.66B. C-SOFTS parameter count grows linearly with channel count because the spatial mixer and spectral modulation scale with channel count. At $C = 5{,}000$, its parameter count is approximately four times larger than that of SOFTS, though this does not meaningfully impact inference time. Notably, peak GPU memory is comparable across all three models, indicating that memory is not a practical differentiator.

In summary, C-SOFTS and I-SOFTS forecasting accuracy gains cannot be attributed to additional compute. Their accuracy gains over SOFTS reflect architectural inductive bias aligned with the dataset's spatial structure. This reinforces the central argument: dataset spatial structure, not model complexity, determines the effective modeling strategy.

## F    Analysis of Learned Spectral Filters

To understand how the frequency mixer component of the Channel Mixer adapts its representation across different multivariate time series, we analyzed the learned complex weight matrix using spectral flatness and rank-1 ratio. Spectral flatness is the ratio of geometric to arithmetic mean power, averaged over the feature dimension $d$. Rank-1 ratio quantifies the similarity of magnitude profiles across feature $d$. We perform singular value decomposition on the magnitude matrix $\mathbf{W} \in \mathbb{C}^{K \times d}$ and compute the proportion of total variance captured by the first singular value: $S_0 / \sum_i S_i$.

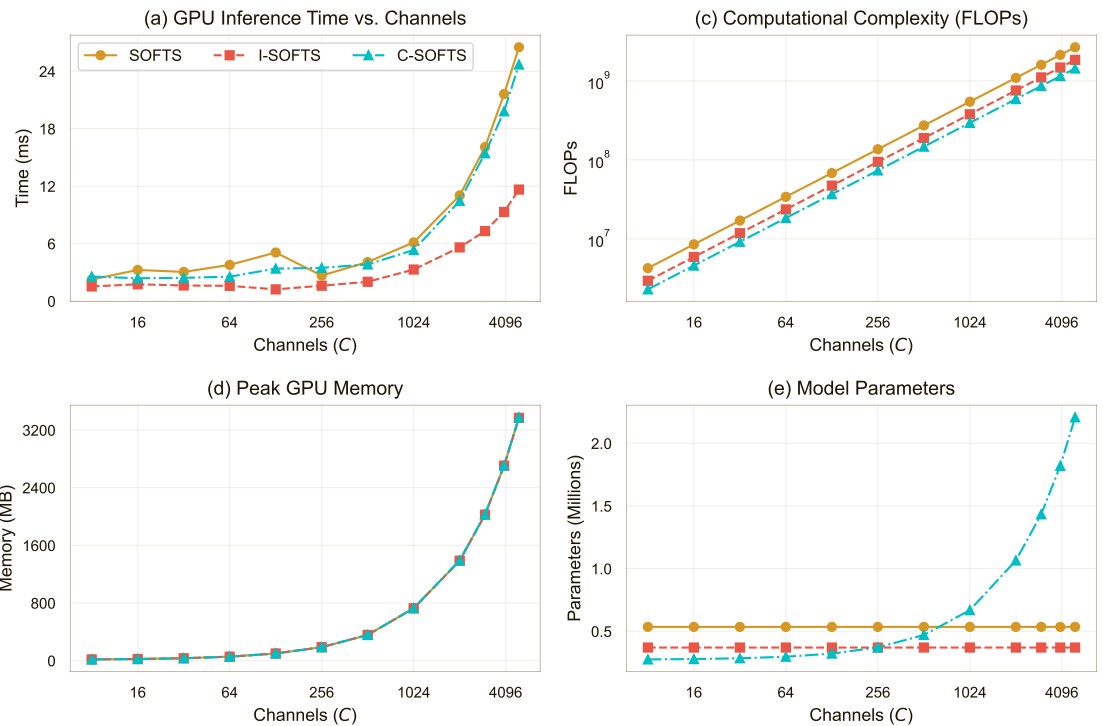

Figure 8: Empirical scaling of SOFTS, C-SOFTS, and I-SOFTS across channel counts $C \in [8, 5000]$. We set the lookback window $L = 96$, horizon $H = 720$, bottleneck dimension $r = 32$, and batch size to 4. I-SOFTS is the most efficient. C-SOFTS parameter count grows linearly with $C$, however it reduces FLOPs at every scale, and has comparable inference time to SOFTS. Peak GPU memory is consistent across all models.

Table 13 summarizes these metrics across datasets grouped by channel count. Figures 9 and 10 visualize the learned magnitude profiles.

For datasets with many channels ($C \geq 137$), the learned filters exhibit near uniform magnitude spectra (spectral flatness > 0.99) and highly diverse feature-specific profiles (rank-1 ratio 0.10-0.21).

For datasets with few channels ($C = 7$), spectral flatness decreases to 0.88-0.97, rank-1 ratios increase to 0.50-0.56, and phase circular variance drops to 0.65-0.76. Within the few-channel group, the effect of forecast horizon varies with temporal resolution. For the hourly ETTh1/2 datasets, longer horizons accentuate low-frequency emphasis, as spectral flatness decreases. For 15-minute ETTm1/2 datasets, the spectrum remains largely flat across horizons. This indicates that the shift towards more structured low-pass filters when tasked with longer prediction is mediated by the data's intrinsic timescales. Weather ($C = 21$) exhibits intermediate characteristics with spectral flatness of 0.93 and a rank-1 ratio of 0.40.

Generally, the gates are soft with mean values around 0.5 across all frequency modes for all datasets, indicating that no frequency band is fully suppressed or saturated. These observations show that the frequency mixer learns qualitatively different filter structures depending on the number of input channels. For larger channels, it converges to a uniform all-pass filter, whilst for smaller channels it shows partial regularization with a higher rank-1 ratio (0.40).

Table 13: Summary of learned spectral filter characteristics across dataset groups. For each dataset, metrics were first averaged over all forecasting horizons. Within each group, the mean and standard deviation were then computed.

| Group | Dataset | Spectral Flatness | Rank-1 Ratio |
| --- | --- | --- | --- |
| Many-channels ($C \geq 137$) | PEMS datasets, ECL, Solar, Traffic | $0.993 \pm 0.003$ | $0.144 \pm 0.041$ |
| Few-channels ($C \leq 21$) | ETT datasets, Weather | $0.939 \pm 0.038$ | $0.498 \pm 0.059$ |

## G   Error Bar

Here, we show the robustness of C-SOFTS and I-SOFTS in Tables 15 and 14, respectively.

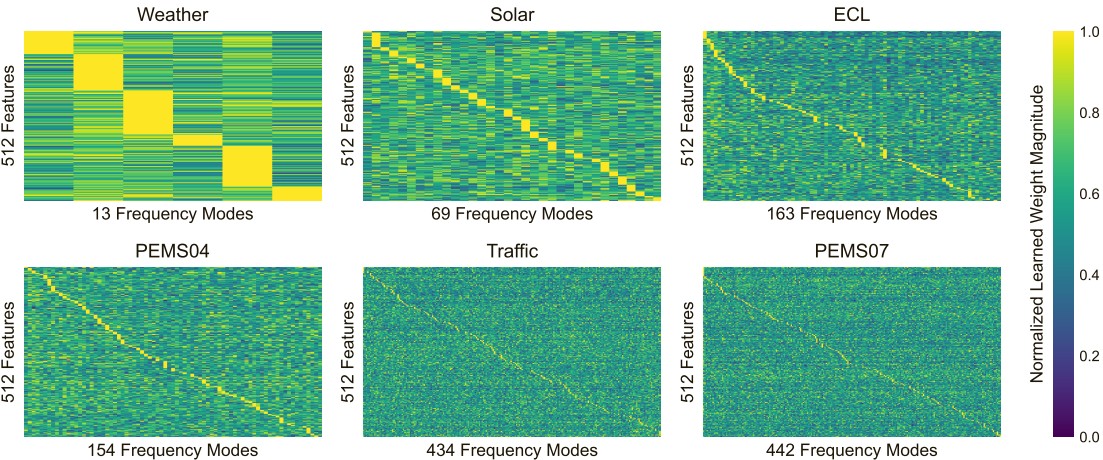

Figure 9: Heatmaps of normalized spectral filter magnitudes for the first encoder layer ($L = 96$, $H = 96$). Features ($d_{\mathrm{model}}$) are sorted by peak frequency. Features and frequency modes are downsampled for visualization.

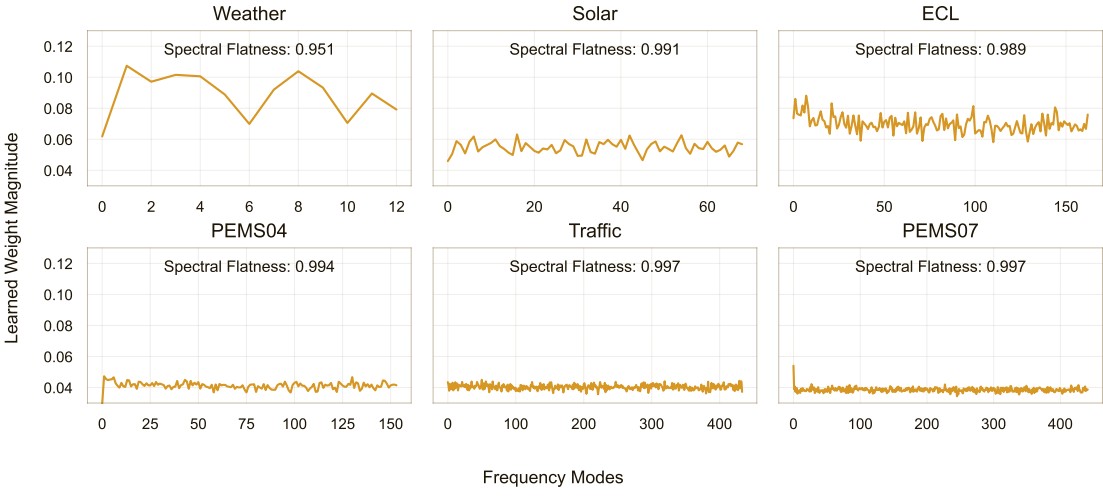

Figure 10: Magnitude and spectral flatness of learned complex scaling weights across spatial frequency modes for the first encoder layer ($L = 96$, $H = 96$). The pattern is consistent across horizons: datasets with larger channel counts converge to a flat magnitude distribution across frequency modes, indicating uniform frequency weighting rather than selective filtering.

Table 14: The robustness of C-SOFTS is evaluated by averaging results over three different random seeds.

| Dataset | ECL | | Solar | | Weather | |
|---|---|---|---|---|---|---|
| Horizon | MSE | MAE | MSE | MAE | MSE | MAE |
| 96 | $0.142 \pm 0.002$ | $0.241 \pm 0.003$ | $0.198 \pm 0.006$ | $0.239 \pm 0.009$ | $0.162 \pm 0.003$ | $0.208 \pm 0.003$ |
| 192 | $0.160 \pm 0.004$ | $0.259 \pm 0.005$ | $0.227 \pm 0.006$ | $0.262 \pm 0.005$ | $0.215 \pm 0.003$ | $0.257 \pm 0.004$ |
| 336 | $0.175 \pm 0.002$ | $0.276 \pm 0.003$ | $0.254 \pm 0.011$ | $0.284 \pm 0.003$ | $0.271 \pm 0.003$ | $0.295 \pm 0.003$ |
| 720 | $0.193 \pm 0.002$ | $0.295 \pm 0.002$ | $0.243 \pm 0.002$ | $0.275 \pm 0.001$ | $0.351 \pm 0.004$ | $0.344 \pm 0.002$ |
| Dataset | PEMS04 | | PEMS07 | | PEMS08 | |
| Horizon | MSE | MAE | MSE | MAE | MSE | MAE |
| 12 | $0.067 \pm 0.001$ | $0.165 \pm 0.002$ | $0.064 \pm 0.002$ | $0.146 \pm 0.001$ | $0.072 \pm 0.001$ | $0.172 \pm 0.003$ |
| 24 | $0.076 \pm 0.001$ | $0.176 \pm 0.002$ | $0.078 \pm 0.003$ | $0.158 \pm 0.001$ | $0.097 \pm 0.002$ | $0.195 \pm 0.003$ |
| 48 | $0.090 \pm 0.002$ | $0.198 \pm 0.005$ | $0.098 \pm 0.001$ | $0.182 \pm 0.005$ | $0.143 \pm 0.002$ | $0.226 \pm 0.003$ |
| 96 | $0.107 \pm 0.002$ | $0.217 \pm 0.002$ | $0.111 \pm 0.004$ | $0.192 \pm 0.001$ | $0.223 \pm 0.011$ | $0.248 \pm 0.014$ |

Table 15: The robustness of I-SOFTS is evaluated by averaging results over five different random seeds.

| Dataset | ETTm1 | | ETTm2 | |
|---|---|---|---|---|
| Horizon | MSE | MAE | MSE | MAE |
| 96 | $0.327 \pm 0.003$ | $0.365 \pm 0.003$ | $0.178 \pm 0.001$ | $0.260 \pm 0.001$ |
| 192 | $0.374 \pm 0.002$ | $0.388 \pm 0.001$ | $0.244 \pm 0.001$ | $0.304 \pm 0.001$ |
| 336 | $0.404 \pm 0.002$ | $0.409 \pm 0.001$ | $0.303 \pm 0.004$ | $0.340 \pm 0.005$ |
| 720 | $0.469 \pm 0.003$ | $0.449 \pm 0.001$ | $0.401 \pm 0.005$ | $0.399 \pm 0.003$ |
| Dataset | ETTh1 | | ETTh2 | |
| Horizon | MSE | MAE | MSE | MAE |
| 96 | $0.378 \pm 0.003$ | $0.397 \pm 0.002$ | $0.297 \pm 0.003$ | $0.346 \pm 0.002$ |
| 192 | $0.435 \pm 0.003$ | $0.428 \pm 0.001$ | $0.374 \pm 0.003$ | $0.396 \pm 0.002$ |
| 336 | $0.482 \pm 0.002$ | $0.454 \pm 0.002$ | $0.421 \pm 0.003$ | $0.432 \pm 0.001$ |
| 720 | $0.508 \pm 0.014$ | $0.491 \pm 0.008$ | $0.427 \pm 0.001$ | $0.445 \pm 0.001$ |

