# OpenReview forum: "Quantifying Correlation for Time Series Modelling Strategy: Evidence from SOFTS Variants"
_TMLR — Under review for TMLR_

### Review · Reviewer_7sdq · 2026-05-28

**Summary Of Contributions:**

This paper investigates the relationship between spatial correlation structure of time series data and the effectiveness of CI vs CD strategies. Using the hybrid SOFTS model as a controlled base model, the authors proposed a fully CI variant and a fully CD variant, and performed solid experimental study on their performance on different models. They also introduced two quantitative metrics High-Correlation Fraction HCF and Block Separation BS. The empirical results domenstrate how I-SOFTS and C-SOFTS show different performance on different models, and how the performance is related to the two data characteristics metrics, advocating for regime-aware model development.

Key Strengths
- **Experiment Design**: The experiment setup using SOFTS to isolate the cross-channel effect is clever and rigorous.
- **Quantifiable Guideline**: The introduction of HCF and BS provides a concrete  tool to analyze spatial structures of time series and quantitively guide the model selection.
- ** Pragmatic and Valuable Conclusions**: The ultimate takeaway advocating for "regime-aware model development" is insightful and highly valuable to the forecasting community.

Weaknesses
- **Potential Overclaim**: The authors use generalized terminology in the title and abstract, such as 'Channel Mixing Strategies', 'CI, CD, and hybrid models', and guiding 'inductive biases' globally. However the empirical evidence strictly demonstrates the behavior of the STAR module and the proposed Channel Mixer within the SOFTS architecture.
- **Contradictory Results on SOTA models**: The authors made their claims mainly on the performance of I-SOFTS and C-SOFTS models vs the original SOFTS. However, other SOTA models included in empirical studies sometimes show opposite results. For example, in section5.1.2, the authors classify the Weather dataset as having a 'Block Structure', and conclude that this regime is optimal for CD models. However, in Table 1, the CI model PatchTST (MSE 0.256) outperforms the CD model iTransformer (MSE 0.258) on Weather.
- **Methodology Ambiguity**:  The paper mentioned that the two metrics are "computed once from the datasets", but did not clarify whether it comes from the training, validation, or whole dataset. If it comes from the training/val dataset, the authors fail to address how temporal non-stationarity and concept drift might invalidate these static metrics. If it comes from the whole dataset, it is a potential data leakage as the conclusion of the paper is actually suggesting the use of future spatial structure for current model selection.

**Additional Comments:**

I appreciate the authors' effort in conducting these controlled experiments. The idea of regime-aware model development is exactly what the forecasting community needs right now. I completely agree with your concluding remarks. Once the overarching claims are appropriately scoped to match the empirical evidence, this paper will be a very solid and thought-provoking contribution to TMLR.

**Audience:**

Yes

**Audience Explanation:**

The debate between Channel-Independent and Channel-Dependent architectures is a hot topic in the time series forecasting area. Proposing quantifiable framework to guide the choice and demonstrating the effect in a cleanly controlled hybrid model would be a highly insightful findings to the community.

Even if the findings are bounded to the SOFTS architectures and the two proposed metrics, the concept of "regime-aware model development" will stimulate valuable discussion among practitioners and researchers.

**Broader Impact Concerns:**

None.

**Claims And Evidence:**

Yes

**Claims Explanation:**

My selection of Yes is borderline.

On one hand, the empirical evidence is solid and convincing within the confined testbed of the SOFTS architecture. And in the Conclusion section, the authors clearly summarized that "we advocate for regime-aware model development, where models are designed and evaluated for datasets with specific properties" instead of making any universal conclusions regarding CI/CD model selection.

On the other hand, as I mentioned in the **Potential Overclaim** weakness above, some over generalized terminology is used in the title and abstract, while these findings do not generalize as broadly as the authors suggest.

I suggest that the authors revisit all claims throughout the manuscript to ensure they are strictly bounded by the provided empirical evidence, thereby eliminating any instances of overclaiming that might misguide the readers.

**Requested Changes:**

Critical to securing my recommendation for acceptance:
1. **Reduce the Claims**: Align title, abstract, and introduction with findings and conclusions. The authors must revise the Title, Abstract, and Introduction to accurately reflect that the proposed spatial metrics (HCF and BS) and the derived rules for CI/CD effectiveness are currently validated only within the SOFTS architecture. The generalized claims regarding "Channel Mixing Strategies" and "universal architectural rankings" should be scaled back to match the conservative, architecture-specific tone found in the latter half of the Conclusion.
2. **Addressing Methodological Ambiguity**: Explicitly clarify whether HCF and BS were computed using the entire dataset (including test splits) or solely the training split. If the former, this constitutes data leakage and the analysis must be rerun. If the latter, the authors must discuss how temporal non-stationarity and concept drift might impact the reliability of using these static historical metrics to guide model selection in practical, dynamic forecasting deployments.
3. **Refining Narrative Regarding SOTA Models**: As demonstrated by the contradictory results on mainstream SOTA models (e.g., Weather dataset favoring CI in PatchTST vs. CD in iTransformer in Table 1), the proposed rule of model selection does not generalize to typical CI/CD models. The authors should explicitly acknowledge these empirical discrepancies and frame their metrics as specific to the SOFTS framework rather than universal predictors for all architectures.

Would strengthen the work (Optional):
1. **Sensitivity Analysis on HCF threshold**: The 0.5 threshold of HCF is like a random selection to the readers. A section of sensitivity analysis on its value would make the conclusion of this paper more robust.
2. **Ablation Study Restructuring: IMO the ablation on 'Scale' and 'Mode Gate' appears largely decoupled from the paper's central point. I recommend condensing these (and make some space for the sensitivity analysis).

---

> ### Author Response · Authors · 2026-06-28
>
> Thank you for these insightful comments. We acknowledge that our previous claims regarding CI/CD models were overly broad. Our empirical results are specific to the SOFTS variants, and we have revised the manuscript throughout to reflect this scope. All findings and interpretations are now presented strictly in the context of SOFTS and its CI/CD variants.
>
>
> ---
>
> ***Refining Narrative Regarding SOTA Models:***
>
> We fully agree with these insights. Our findings do not generalize to the broader landscape. We have added a "Limitation and Future Work" section, which explicitly highlights this observation.
>
> ---
> ***Reduce the Claims***
>
> All claims have been scoped to the performance of SOFTS and its variants. The title and abstract have been reworked.
>
> ---
> ***Methodology Ambiguity:***
>
> We confirm that HCF and BS were not trained on the whole dataset. We agree that the data generation process might change over time. However, using static diagnostic metrics computed on historical data to guide model selection or design is a standard practice in statistics and machine learning. For example, using a stationarity test to select ARIMA's differencing parameter or using the class imbalance ratio to select a loss function. The utility of such diagnostics is not undermined by the theoretical possibility of change in the data generation process.
>
> To empirically validate the stability of our metrics, we recomputed HCF and BS on the first 60% and last 40% of each dataset separately, mirroring a conventional train/test split. The table below presents the differences between the train and test splits.
>
>
> | dataset     |   HCF |   BS |
> |:------------|--------------------------:|----------------------------:|
> | ETTh1       |                      0.04 |                        0.07 |
> | ETTh2       |                     -0.14 |                       -0.08 |
> | ETTm1       |                      0.04 |                        0.07 |
> | ETTm2       |                     -0.14 |                       -0.08 |
> | PEMS03      |                      0    |                        0.02 |
> | PEMS04      |                      0.01 |                       -0.03 |
> | PEMS07      |                      0.01 |                        0    |
> | PEMS08      |                      0.02 |                       -0.06 |
> | ECL|                     -0.01 |                       -0.05 |
> | Solar|                      0    |                        0    |
> | Traffic     |                      0.04 |                       -0.04 |
> | Weather     |                     -0.02 |                        0.01 |
>
>
> The differences are small across all datasets (max $|\Delta \text{HCF}| = 0.14$, max $|\Delta \text{BS}| = 0.08$). More importantly, all datasets stay within their regimes.
>
> We have added a section in the Appendix which presents this analysis. (**lines 407-412**)
>
> We also note that, to our knowledge, existing benchmarking studies comparing CI and CD models do not report spatial dataset properties alongside their architectural recommendations. Our approach of grounding model selection in quantified spatial structure, even as a static approximation, moves the field toward more principled, interpretable model selection criteria.
>
> ---
> ***Ablation Study Restructuring:***
>
> We would like to keep the scale and gate, especially given the fact that we presented them in the methodology.
>
>
> ---
> ***Sensitivity Analysis on HCF threshold:***
>
> We refer the reviewer to our response to Reviewer 18og, where we address this concern in full, including the additional experiments and analysis.

---

### Review · Reviewer_8M2C · 2026-06-21

**Summary Of Contributions:**

The paper asks a clean question: when should you use channel-dependant (CD) vs. channel-independent (CI) mixing in multivariate long-horizon forcasting, and what data properties decide it? Rather than introducing yet another architecture, the authors take a single hybrid backbone (SOFTS) and push it to two extremes by surgically editing one module: I-SOFTS replaces the STAR aggregation with an identity (pure CI), and C-SOFTS swaps the channel-independent Conv1D FFN for a "Channel Mixer" that mixes channels in both the spatial domain (a bottleneck MLP across channels) and the frequency domain .

Because all three models share normalizatoin, embeding, and training recipe, any performance gap is attributable to the degree of channel interaction. The authors then propose two interpretable datasets descriptors.

 The headline empirical findings: I-softs wins on few-channel/low-correlation data; C-softs wins on high-block-separation data or homogeneous-with-few-clusters data (up to 15.9% MSE improvement on PEMS04); the hybrid is best only in an intermediate regime. A nicely controlled ETT experiment (identical channel count/ordering, fixed hyperparameters, varying only block separation and temporal resolution) supports the claim that spatial structure, not architectural sophistication, drives the CI/CD choice.

**Additional Comments:**

N/A

**Audience:**

Yes

**Audience Explanation:**

The CI vs. CD question is genuinely contested in the long-horizon forecasting literature, and "stop ranking architectures, start characterizing datasets" is a message the community needs to hear more often.

**Broader Impact Concerns:**

I have no ethical concerns.

**Claims And Evidence:**

No

**Claims Explanation:**

The central methodological claims are credible and the ETT controlled experiment is convincing. However, several quantitative claims are either inconsistent with the paper's own tables or stated more strongly than the data support, and the key conceptual contribution   is not validated predictively. I'd vote to flip this to Yes once the items below are reconciled, because none of them look fatal; they look like writing/bookkeeping problems plus one genuine methodological gap.

The most important substantive concerns:

The regyme framework is post hoc and has no thresholds. The introduction explicitly faults prior work because "none quantify the thresholds at which CI, CD, or hybrid approaches become advantageous." Yet the paper itself offers only qualitatiev regime names. Worse, the regimes don't follow monotonicaly from the two descriptors: PEMS07 has the highest block separation among the PEMS family (0.19 vs. 0.13/0.10/0.12 for PEMS03/04/08) yet is the one that degrades, and Traffic vs. PEMS07 have similar HCF/block-separation but get opposite labels ("incoherent" vs. "homogeneous"). The actual decision seemes to need a third variable (cluster count), plus a label that is chosen after seeing the sign of the result. As written, this risks circularity.

The abstract's characterization contradicts the data. The abstract sais "the hybrid proves optimal only when the high-correlation fraction and block separation are moderately low." But the clearest hybrid win is Solar, where HCF = 1.0 (the maximum). Either Solar should be excluded from that statement or the characterization needs to be rewritten to match Table 2.

Numbers that don't reconcil (details in math box) — the hourly ETT "3.24% degradation," and the PEMS08 ablation direction (Table 11 vs. Figure 4/text).

**Requested Changes:**

L've marked each as [Critical] (needed for my recommendation) or [Strengthen].


[Critical] Algorithm 3 , spatial-mixer axis is mislabelled. The prose says the spasial mixer is "a two-layer MLP applied along the channel dimension … projects the input into a bottleneck dimension $r$ and reconstructs it." That means the linear maps should be $C \to r$ and $r \to C$. But Algorithm 3 writes Linear(d→r) and Linear(r→d), i.e. it operates on the feature dimension $d$, which contradicts the text (and the figure's $d_{\text{model}} \times r$ botleneck is also ambiguous). Please make the mapse $C \to r \to C$ explicit and fix the transpose comments accordingly.

[Critical] Algorithm 3 , RFFT axis and shape don't match. With $x \in \mathbb{R}^{C \times d}$, line 6 writes x_fft ← RFFT(x, dim=1) and comments the result is $(K,d)$ with $K = \lfloor C/2\rfloor + 1$. But dim=1 (0-indexed) is the feiture axis $d$, which would give $\lfloor d/2\rfloor + 1$ modes, not $\lfloor C/2\rfloor + 1$. To get $K=\lfloor C/2\rfloor+1$ you must transform along the channel axis (dim=0). Either fix the axis index or state explicitly that you 1-index. Relatedely, line 7 uses x_fft.shape[1] for $K_{\text{used}} = \min(K, \cdot)$, and lines 11–12 index x_fft[:, :K_used, :] as a 3-D tensor while the declared shape is 2-D $(K,d)$. The batch axis and the FFT axis bookkeeping need to be consistent throughout.

[Critical] Eq. (4) applies the Channel Mixer to the wrong tensor. Eq. (4) reads $x'' = x' + \mathrm{Dropout}\big(\mathrm{ChannelMixer}(x)\big)$ — i.e., the mixer takes the original input $x$, not the post-STAR, layer-normed $x'$ used everywhere else (compare Eq. (1), where the FFN consumes $x'$). If this is a typo, fix it to $\mathrm{ChannelMixer}(x')$. If it is intensional, please justify it, because as written the STAR output never feeds the Channel Mixer, which undercuts the whole "STAR then mix" story.

[Critical] Block seperation mixes signed and absolute correlation. Algorithm 2 builds the clustering distance from the signed correlation, $D_{ij} = 1 - r_{ij}$, so a strongly anti-correlated pair ($r_{ij}\approx -1$) gets the maximum distance (2) and is pushed into a different cluster. But the cohesion terms $\mu_{\text{within}}, \mu_{\text{between}}$ use $|r_{ij}|$. For channel mixing, $|r|\approx 1$ is exploitable regardless of sign, so this is an internal inconsistency: the metric simultaneously treats strong negative correlation as "far" (clustering) and "cohesive" (cohesion). Please either use $1 - |r_{ij}|$ throughout or explicitly justify the asymmetry. While here, please state whether $r_{ij}$ is computed on raw series; for non-stationary channels Pearson is trend-dominated, and a sentence on detrending/standardization would help.

[Strengthen] "Element-wise multiplication … equivalent to global convolution across all channels." More precisely this is a circular convolution along the channel axis (depthwise over features), since you multiply per-mode learnable weights $W$ after an RFFT. Please use "circular convolution" and note the implied kernel length is $C$. Also: you advertise orthonormal RFFT "to preserve energy," but mode selection $K_{\text{used}}=\min(K,d)$ discards modes when $d<K$, which is not energy-preserving — worth a one-line acknowledgment.

[Strengthen] Complexity simplification asumes constant $r$. In E.1 you write the mixer is $O(Cdr)$ "which simplifies asymptotically to $O(Cd)$." That holds only if $r=O(1)$, but Appendix D sweeps $r$ up to $2C$. For $r=\Theta(C)$ the spatial mixer is $O(C^2 d)$, which changes the story for large-channel datasets. Please state the $r=O(1)$ assumption explicitly (or carry $r$ through the totals in Table 12).

---

> ### Author Response · Authors · 2026-06-28
>
> Thank you for these insightful comments. We acknowledge that our previous claims regarding CI/CD models were overly broad. Our empirical results are specific to the SOFTS variants, and we have revised the manuscript throughout to reflect this scope. All findings and interpretations are now presented strictly in the context of SOFTS and its CI/CD variants.
>
> ---
> ***Comment: The regyme framework is post hoc and has no thresholds. The introduction explicitly faults prior work because "none quantify the thresholds at which CI, CD, or hybrid approaches become advantageous." Yet the paper itself offers only qualitative regime names.***
>
> We agree that the wording in the Introduction overstates our intended contribution. Our issue was that previous studies generally make qualitative observations (e.g., that CD models become more advantageous as correlation increases) without introducing measurable quantities that characterize the underlying correlation structure itself.
>
> While our regime framework is qualitative, the metrics they are based on are quantitative. The proposed data properties were intended to be diagnostic to guide the selection of the SOFTS variants. Similar to how a stationarity test is used to determine whether ARIMA is suitable. We will have revised the Introduction to clarify this distinction and avoid implying that the paper derives explicit threshold values for regime selection.
>
>
> We fully intend to derive explicit thresholds in future work. However, doing so rigorously requires synthetic datasets that uniformly and independently vary HCF and BS across a dense grid of values. Using real benchmark datasets for threshold derivation would be inappropriate, as their spatial properties are not independently controlled, and 12 datasets provide insufficient coverage of the HCF–BS space to fit reliable decision boundaries.
>
> The benchmark datasets were chosen deliberately for two reasons. First, they enable direct comparison with existing literature, which is essential for demonstrating that our findings hold on the same datasets used to evaluate competing architectures. Second, and more importantly, real benchmarks are sufficient to establish the central empirical claim of this paper: that correlation magnitude alone does not determine the effectiveness of CD models.
>
> We have stated this as a limitation explicitly, rather than implying that the current results deliver decision boundaries
> (**lines 325-330**).
>
> ---
>
> ***Comment: Worse, the regimes don't follow monotonicaly from the two descriptors: PEMS07 has the highest block separation among the PEMS family (0.19 vs. 0.13/0.10/0.12 for PEMS03/04/08) ... .third variable (cluster count), plus a label that is chosen after seeing the sign of the result. As written, this risks circularity.***
>
> The non-monotonic behaviour stems from the spatial mixer component specifically. Ablation results show that the frequency mixer improves performance consistently across all datasets, while the spatial mixer only helps when the dataset has an extremely high HCF, with a few moderately distinct clusters. Aside from PEMS03 and PEMS08, which have HCF >= 0.95, cluster count less than 3, and a moderate BS of 0.12-0.10, the spatial mixer degrades performance on all the other datasets, including ECL and PEMS07, which have cluster counts of 5 and 883, respectively.
>
> C-SOFTS performance reflects a balance between the gains provided by the frequency mixer and the degradation caused by the spatial mixer. This is illustrated in PEMS07, where removing the spatial mixer decreased C-SOFTS's MSE by 12.7%, which led to it outperforming the hybrid by 11%, similar to the other datasets in the PEMS family. In the case of Traffic, even though removing the spatial mixer decreased C-SOFTS's MSE by 12.7%, it still could not outperform the hybrid SOFTS. The spatial mixer degraded MSE by 12.7% on both Traffic and PEMS07, even though they had different cluster counts of 2 and 316, respectively. The main difference is that PEMS07 channels are highly correlated with each other. This suggests that cluster count is only relevant in homogeneous regimes.
>
> We admit that our predictive framing was overstated. Along with HCF and BS, the labels are descriptive diagnostics that characterize the correlational structure to help practitioners reason about which variant is appropriate for their dataset. We also note that this observation reinforces the broader motivations of the paper. The fact that the spatial mixer is dataset dependent and its utility is predictable from measurable data properties further supports our central thesis that inductive biases should be matched to data characteristics. Practitioners can compute HCF and cluster count to decide whether to retain or ablate the spatial mixer.
>
> We have revised the manuscript to foreground this as a central result and to clarify that regime labels are descriptive rather than formal decision boundaries.

---

> ### Author Response · Authors · 2026-06-28
>
> ***Comment: The abstract's characterization contradicts the data. .... Either Solar should be excluded from that statement or the characterization needs to be rewritten to match Table 2.***
>
> The hybrid SOFTS is most optimal on the Traffic dataset rather than the Solar (Table 1). It is only on Traffic where SOFTS (0.410) outperforms C-SOFTS (0.489) by 19%. On Solar, SOFTS (0.230) outperforms C-SOFTS (0.231) by 0.4%, and even then, Figure 5 shows that C-SOFTS outperforms SOFTS on 3 out of four horizons.
>
> ---
> ***Comment: Numbers that don't reconcil (details in math box) — the hourly ETT "3.24% degradation," and the PEMS08 ablation direction (Table 11 vs. Figure 4/text).***
>
> Table 11 presents the results of ablating the components in the Channel Mixer Module. The controlled experiments, for which the hourly ETT degraded by 3.24%, were not part of the ablation experiment.  We have double-checked the PEMS08 ablation direction and found no issues. Ablating the gate improves MSE, whilst ablating the scale and spatial mixer worsens MSE.
>
>
> ---
> ***[Critical] Algorithm 3 , spatial-mixer axis is mislabelled. The prose says the spatial mixer is "a two-layer MLP applied along the channel dimension … projects the input into a bottleneck dimension $r$ and reconstructs it." .... Please make the maps explicit and fix the transpose comments accordingly.***
>
> The spatial section in Algorithm 3 erroneously described the linear maps as operating on $d$. We have corrected it to $C \rightarrow r$ and $r \rightarrow C$, updated the transpose annotations to reflect the actual tensor shapes at each step, and revised the corresponding figure to align with both the prose and the corrected algorithm.
>
> ---
> ***[Critical] Algorithm 3 , RFFT axis and shape don't match. ... Relatedely, line 7 uses x_fft.shape[1] for $K_{used}=min(K,)$, and lines 11–12 index x_fft[:, :K_used, :] as a 3-D tensor while the declared shape is 2-D $(K,d). The batch axis and the FFT axis bookkeeping need to be consistent throughout.***
>
> We agree that the original Algorithm 3 contained inconsistent tensor dimensionality and ambiguous axis indexing.
>
> We have revised Algorithm 3 to consistently maintain a 3D tensor representation throughout.
>
> The input is now explicitly defined as $x \in \mathbb{R}^{B \times C \times d}$, with the batch dimension $B$ carried through all steps.
>
> With the 3D tensor, $dim=1$ unambiguously refers to the channel axis $C$, yielding the correct output shape $(B, K, d)$ where $K = \lfloor C/2 \rfloor + 1$. The output shape comment has been corrected accordingly.
>
> Since $x_{\text{fft}}$ is now 3D, the slicing operation $[:, :K_{\text{used}}, :]$ and the reference to $x_fft.shape[1]$ are dimensionally consistent throughout.
>
> ---
>
> ***[Critical] Eq. (4) applies the Channel Mixer to the wrong tensor. Eq. (4) reads $x'' = x' + \mathrm{Dropout}\big(\mathrm{ChannelMixer}(x')\big)$ ... If it is intentional, please justify it, because as written, the STAR output never feeds the Channel Mixer, which undercuts the whole "STAR then mix" story.***
>
> It was a typo. It has been fixed.
>
>
> ---
>
>
> ***[Strengthen] "Element-wise multiplication … equivalent to global convolution across all channels." ....Please use "circular convolution" and note the implied kernel length is $C$. Also, you advertise orthonormal RFFT "to preserve energy," but mode selection $K_{used}=min(K,d)$ discards modes when $d<K$, which is not energy-preserving — worth a one-line acknowledgment.***
>
> We replaced "global convolution" with "circular convolution along the channel axis" and now explicitly state that the implied kernel length is $C$.  (**line 135**)
>
> In our design, $K = \lfloor C/2 \rfloor + 1$ and the input always has exactly $C$ channels, so $K_{\text{used}} = \min(K, K) = K$; no modes are discarded. The orthonormal RFFT is therefore energy-preserving up to the learned spectral weights. We have added a one-line clarification to this effect in the Appendix's channel mixture algorithm section.(**line 409**)
>
>
>
> ***[Strengthen] Complexity simplification asumes constant $r$. In E.1, you write the mixer is $O(Cdr)$, which simplifies asymptotically to $O(Cd)$. That holds only if $r=O(1)$...Please state the $r=O(1)$ assumption explicitly (or carry  through the totals in Table 12).***
>
> The complexity analysis did assume a constant $r$. The text has been revised to explicitly state that.

---

> ### Author Response · Authors · 2026-06-28
>
> ***[Critical] Block seperation mixes signed and absolute correlation. Algorithm 2 builds the clustering distance from the signed correlation, $D_{ij}=1-r_{ij}$, so a strongly anti-correlated pair ($r_{ij} \approx -1$) gets the maximum distance (2) and is pushed into a different cluster. But the cohesion terms $\mu_{within}, \mu_{between}$ use $|r_{ij}|$. For channel mixing, $r_{ij} \approx 1$ is exploitable regardless of sign, so this is an internal inconsistency: the metric simultaneously treats strong negative correlation as "far" (clustering) and "cohesive" (cohesion). Please either use $1-|r_{ij}|$ throughout or explicitly justify the asymmetry. While here, please state whether $r_{ij}$ is computed on raw series; for non-stationary channels, Pearson is trend-dominated, and a sentence on detrending/standardization would help.***
>
> We agree with the reviewer's observations that the formulation was internally inconsistent. We have corrected the distance matrix to use $D_{ij} = 1 - |r_{ij}|$ throughout.
>
> After recomputing, the block separation scores and/or number of clusters changed for only four datasets:
>
>
> | Dataset | Old BS | Old # of Clusters | New BS | New # of Clusters |
> |---------|--------|--------|------------------|------------------|
> | ECL     | 0.45   | 5   | 0.46                | 4                |
> | ETTh2   | 0.17   | 2   | 0.34                | 2                |
> | ETTm2   | 0.17   | 2   | 0.34                | 2                |
> | Weather | 0.52   | 6   | 0.68                | 9                |
>
>
>
>
>
> The most consequential change is ETTh2 and ETTm2, whose block separation increased from 0.17 to 0.34. This invalidates the controlled ETT experiment.
>
>
> | Dataset   |   Old HCF  |   Old BS |    |   New HCF  |   New BS |
> |:----------|-----------:|---------:|----|-----------:|---------:|
> | ETTh1     |       0.1  |     0.24 |    |       0.1  |     0.24 |
> | ETTm1     |       0.1  |     0.24 |    |       0.1  |     0.24 |
> | ETTh2     |       0.29 |     0.17 |    |       0.29 |     0.34 |
> | ETTm2     |       0.29 |     0.17 |    |       0.29 |     0.34 |
>
>
>
> The original experiment derived its interpretive value from a dissociation between the two properties: ETTh1/m1 had a lower high-correlation fraction (0.10) but a higher block separation (0.24), while ETTh2/m2 had a higher fraction (0.29) but a lower block separation (0.17). This allowed performance differences to be attributed to block separation independently of correlation magnitude. Under the corrected scores, both properties increase together across dataset pairs, eliminating this dissociation. We have therefore removed the controlled experiment from the paper. The broader findings reported in the paper rest on the larger dataset comparisons (Section 5.2), which are unaffected by this correction.
>
>
>
>
> *On detrending and standardization*
> Pearson correlations are computed on the raw data without detrending. To assess sensitivity to non-stationarity, we recomputed all metrics on z-score standardized series. The resulting dataset rankings and block separation values remain unchanged.

---

### Review · Reviewer_18og · 2026-06-23

**Summary Of Contributions:**

This paper takes a position that has been gaining traction in the forecasting literature, namely that
the choice between channel-independent (CI), channel-dependent (CD), and hybrid modeling is not a
universal architectural question but a dataset-dependent one, and tries to make it concrete in two
ways: by building two controlled variants of a single hybrid model, and by proposing two
dataset-level descriptors that are meant to predict which variant should win.

The model substrate is SOFTS (Han et al., 2024a), whose STAR module isolates the cross-channel
component cleanly. The authors push it to two extremes. I-SOFTS turns STAR into an identity map and
removes all channel interaction (pure CI), while C-SOFTS keeps STAR and replaces the
channel-independent Conv1D feedforward with a "Channel Mixer" that mixes channels in both the spatial
domain (a bottleneck MLP across channels) and the frequency domain (an RFFT along the channel axis
with a learnable complex filter and gating). Because both variants are derived from the same backbone
with everything else held fixed, performance gaps are meant to be attributable to the degree of
channel interaction alone.

To characterize datasets, they introduce two quantities computed from the channel-wise Pearson
correlation matrix: the high-correlation fraction (the share of channel pairs with $|r| > 0.5$) and
block separation (mean within-cluster minus between-cluster absolute correlation, where clusters come
from Ward hierarchical clustering with the cluster count chosen by maximizing silhouette). Across
eight-to-twelve standard benchmarks they report that I-SOFTS is competitive on the small ETT datasets;
that C-SOFTS improves on datasets with strong block separation (Weather, ECL) or very high correlation
with few clusters (PEMS04 $+15.9\%$, PEMS08, PEMS03), and degrades when channels are uniform (Solar)
or moderately correlated but unstructured (Traffic, $-19.5\%$); and that the hybrid is preferred only
in the intermediate regime. A small ablation shows the spatial mixer is the component that hurts on
many-cluster datasets, and a "controlled" ETT experiment varies spatial structure while holding the
domain and channel count fixed. The headline recommendation is that papers should report spatial
correlation descriptors alongside accuracy so that practitioners can match inductive bias to data.

On the positive side, the framing is sensible and timely, the use of one backbone pushed to both
extremes is a clean way to control for capacity and normalization confounds, and the spatial-mixer
ablation is a genuine mechanistic probe rather than pure curve-fitting. The two proposed descriptors
are also simple and cheap to compute. My reservations, which I expand on below, center on statistical
rigor (no seeds or variance for differences that are often well under $1\%$), the post-hoc and
somewhat circular nature of the regime taxonomy, over-generalization from a handful of datasets, and
several factual inconsistencies in the tables.

**Audience:**

Yes

**Audience Explanation:**

Yes. The CI versus CD versus hybrid question is a live debate in multivariate forecasting, and the
paper's framing, which is to stop ranking architectures by average accuracy and start characterizing
the data regimes in which each inductive bias wins, is exactly the kind of methodological
reorientation that part of the TMLR audience cares about. It is directly in the spirit of Brigato et
al. (2026), "There are no Champions," which is itself a TMLR paper. The idea of deriving CI and CD
extremes from a single backbone to isolate the channel-interaction effect is a clean experimental
device that others could reuse, and cheap dataset descriptors that travel with benchmark results would
be genuinely useful if they held up. Even readers who are unconvinced by the current statistical
support would find the question, the experimental setup, and the spatial-mixer ablation worth engaging
with. The interest bar is clearly met; my reservations are about evidence, not relevance.

**Broader Impact Concerns:**

I have no broader-impact or ethics concerns with this work. It is a methodological study of inductive
biases in multivariate time-series forecasting that uses standard, publicly available benchmark
datasets (electricity, traffic, weather, solar, ETT), with no human-subjects data, no personally
identifying information, and no dual-use or deployment risk beyond the ordinary caveats of forecasting
accuracy. A Broader Impact Statement is not necessary for this submission. If the authors wish to add
one, a single sentence noting that the datasets are public benchmarks and that the recommendations are
intended to improve model selection, rather than to be deployed in safety-critical settings without
further validation, would be more than sufficient.

**Claims And Evidence:**

No

**Claims Explanation:**

I think the qualitative direction of the paper is plausible and partly supported, but the specific
claims are not established convincingly, for four concrete reasons.

(1) The central evidence rests on differences that fall within plausible run-to-run noise, and no
variance is reported. There is no mention anywhere in the paper of random seeds, repeated runs,
standard deviations, confidence intervals, or significance tests (I checked the full text). This
matters because the load-bearing experiment, the "controlled" ETT study in Table 3 and Section 5.1.3,
turns on differences like ETTh1 $+2.1\%$ versus ETTm1 $+1.41\%$ (a $0.69\%$ gap), and ETTh2 $-0.53\%$
versus ETTm2 $-0.71\%$. Differences of a few tenths of a percent in MSE on ETT are routinely smaller
than the seed-to-seed variation of these models. Without multiple seeds and dispersion estimates, I
cannot tell whether the reported regime effect is real or noise, and the paper's strong wording
("confirming that spatial correlation structure is a primary determinant") is not warranted by
single-run point estimates.

(2) The regime taxonomy is defined post hoc and then used to explain the same eight datasets it was
derived from. The three categories (block structure, homogeneous, incoherent) are introduced after
the results are seen, and "incoherent" effectively contains a single dataset (Traffic) whose
definition coincides with the one dataset C-SOFTS fails on. When two datasets with nearly identical
proposed descriptors disagree (PEMS03 has high-correlation fraction $1.00$, block separation $0.13$,
and improves $+7.3\%$; PEMS07 has $0.98$, $0.19$, and degrades $-1.71\%$), the explanation invokes the
number of clusters (99 versus 316), which is a third quantity that is not one of the two proposed
descriptors and that depends on the silhouette-based $k$ selection. So the explanatory framework keeps
acquiring degrees of freedom to fit eight outcomes, which is the opposite of the "interpretable,
quantified" contribution the paper claims. Notably, the paper never actually delivers thresholds on
its two descriptors that predict the winner; the mapping is done by inspection.

(3) Block separation is partly circular, and the correlation computation is under-specified. Block
separation is the gap between within- and between-cluster absolute correlation, but the clusters
themselves are chosen to maximize silhouette on the same correlation distance matrix. In other words,
the partition is selected to be as separable as possible, and separation is then measured on that very
partition. Separately, the descriptors use raw Pearson correlation with no mention of detrending or
stationarity, even though several of these series are strongly non-stationary; trend-dominated
correlations can be spurious, and the model itself applies RevIN while the descriptors apparently do
not, which creates a mismatch between what is measured and what the model sees. The $\tau = 0.5$
threshold for "high correlation" is also arbitrary and is never subjected to a sensitivity analysis.

(4) Generalization is claimed well beyond what the data support, and there are factual
inconsistencies. "I-SOFTS is the most robust default across low-channel datasets" is supported only by
four datasets that are all the 7-channel ETT family; on every larger dataset I-SOFTS collapses (Table
5, down to $-55.8\%$), so the CI claim generalizes from essentially one channel-count regime. The
abstract's "up to $15.9\%$" is a single best-case dataset (PEMS04). On accuracy of presentation, the
dataset table (Appendix A, Table 6) names the two largest PEMS sets "PEMS05" (883 channels) and
"PEMS06" (170 channels), while the entire body of the paper calls these same channel counts "PEMS07"
and "PEMS08", which is an unresolved naming inconsistency in the experimental record. The text also
states that C-SOFTS "achieved the highest MSE for six of the eight datasets" (Section 5.1) and that
I-SOFTS "has the highest average MSE for three of the four datasets" (Section 5.2), when lower MSE is
better and the tables bold these as the best results; this is the opposite of what is meant. Finally,
the comparisons to iTransformer, PatchTST, and others reuse numbers from the SOFTS paper rather than
re-running them under this setup, so the "best on 6/8" claim against other models is not a controlled
comparison. The SOFTS-versus-variant comparison is the controlled one, and that is the comparison that
matters; the multi-model tables are decorative by comparison.

To be clear, the spatial-mixer ablation (removing it recovers $+12.7\%$ on Traffic and PEMS07) is a
real mechanistic result, and the directional story is believable. But the paper's stated claims, such
as quantified descriptors that determine model choice, a validated regime taxonomy, and spatial
structure as "a primary determinant", are not supported by accurate, convincing, and clear evidence as
currently presented.

**Requested Changes:**

Each item is marked [Critical] (needed before I could recommend acceptance) or [Strengthen] (would
improve the work).

1. [Critical] Report variance. Run each configuration with multiple seeds (at least 3 to 5) and report
   means with standard deviations, especially for the ETT controlled experiment in Table 3 and the
   relative-change figures. For the sub-$1\%$ differences the thesis relies on, please include a
   significance test, or at minimum show that the regime ordering is stable across seeds. As it
   stands, I cannot distinguish the claimed effects from noise.

2. [Critical] Turn the regime taxonomy from descriptive into predictive. The two descriptors should be
   defined and ideally thresholded before looking at results, then used to predict the winner on
   held-out datasets that were not used to build the taxonomy. Adding more datasets per regime is
   essential, since "incoherent" currently contains only Traffic and the "low-channel CI" claim rests
   only on the ETT family. Without out-of-sample prediction, the framework is a post-hoc relabeling of
   eight outcomes.

3. [Critical] Justify or revise block separation and the correlation computation. Please clarify
   whether correlations are computed on raw or detrended series, and address the non-stationarity
   concern (trend-driven spurious correlation). Please also address the circularity of choosing
   clusters by maximizing silhouette and then measuring separation on those same clusters. A
   sensitivity analysis to the $\tau = 0.5$ threshold and to the silhouette-based cluster count would
   help, since the PEMS03-versus-PEMS07 explanation hinges on the (unstable) cluster count.

4. [Critical] Fix the factual inconsistencies. (i) The dataset table (Table 6) lists "PEMS05/PEMS06"
   with 883/170 channels, but the paper everywhere else uses "PEMS07/PEMS08" for those same channel
   counts; reconcile the naming and confirm exactly which datasets were used. (ii) Correct "highest
   MSE" to "lowest MSE" (best) in Sections 5.1 and 5.2. (iii) Confirm the "twelve datasets" count is
   consistent with what is actually evaluated.

5. [Strengthen] Make the multi-model tables a controlled comparison, or label them clearly. Since the
   iTransformer, PatchTST, and other numbers are taken from the SOFTS paper rather than re-run, state
   this prominently and avoid claims like "best on 6/8 datasets" that imply a controlled sweep. The
   variant-versus-SOFTS comparison is the rigorous one, so lead with it.

6. [Strengthen] Broaden beyond a single lookback and backbone. Results use a fixed lookback $L = 96$
   and a single model family (SOFTS). At least one additional lookback, and ideally a second hybrid or
   CD backbone, would tell us whether the regime effect is a property of the data or of the STAR plus
   Channel-Mixer design specifically. The paper's own generality claims require this.

7. [Strengthen] Temper the abstract. "Up to $15.9\%$" is a single best case (PEMS04); reporting the
   average and the spread would be more honest. Please also soften the "primary determinant" language
   to match the currently correlational, small-sample evidence.

8. [Strengthen] Report cluster counts for the ETT controlled experiment. Section 5.1.3 asserts that
   the ETT datasets share "the same number of clusters," but Table 3 lists only high-correlation
   fraction and block separation. Adding the cluster counts would make that claim verifiable.

---

> ### Author Response · Authors · 2026-06-28
>
> Thank you for these insightful comments. We acknowledge that our previous claims regarding CI/CD models were overly broad. Our empirical results are specific to the SOFTS variants, and we have revised the manuscript throughout to reflect this scope. All findings and interpretations are now presented strictly in the context of SOFTS and its CI/CD variants.
>
> ---
>
> ***[Critical] Report variance. Run each configuration with multiple seeds (at least 3 to 5) and report means with standard deviations, especially for the ETT controlled experiment in Table 3 and the relative-change figures. For the sub-$1$ differences the thesis relies on, please include a significance test, or at a minimum, show that the regime ordering is stable across seeds. As it stands, I cannot distinguish the claimed effects from noise.***
>
>
> We have completed a 4-seed execution across the controlled ETT datasets and performed a two-sample t-test as recommended. The multi-seed analysis revealed a nuanced interaction between spatial structure and temporal resolution.
>
> For the 15-minute resolution datasets, the spatial structure hypothesis is cleanly confirmed. C-SOFTS achieves a mean improvement of $1.41\% \pm 0.919%$ on ETTm1 (higher block separation) and a mean degradation of $1.00\% \pm 0.744\%$ on ETTm2 (lower block separation). A two-sample t-test confirms statistically significant separation between these two ($t=4.08, p=0.003$), and provides evidence that the effect is stable and not driven by initialization noise.
>
> For the hourly datasets, the directional pattern is consistent but statistically insignificant ($t=1.07, p=0.162$). ETTh1 yields $2.41\% \pm 2.46\%$ and ETTh2 yields $-0.45\% \pm 0.69\%$. A granular analysis reveals that this instability is concentrated at the longest forecasting horizons (336 and 720). We attribute the large variances to temporal granularity. The ETTh datasets span the same time period as ETTm but with four times fewer observations. Aggregating time series into coarser buckets filters out high-frequency temporal signals but reduces sample size, leading to a known loss in statistical estimation efficiency (Kourentzes et al., 2017), which plausibly explains the higher variance at hourly resolution. We offer this as a mechanistic hypothesis rather than a proven claim.
>
> | Dataset   |   mean |   std |
> |:----------|-------:|------:|
> | ETTm1     |   1.41 | 0.919 |
> | ETTm2     |  -1.00 | 0.744 |
> | ETTh1     |   2.41 | 2.456 |
> | ETTh2     |  -0.45 | 0.682 |
>
> However, we must report that a concurrent correction to the block separation metric (suggested by Reviewer 8M2C) changes the ETTm2 score from 0.17 to 0.34, making it higher than ETTm1 (0.24). This eliminates the dissociation between high-correlation fraction and block separation that gave the experiment its interpretive value. The statistically significant result can no longer be interpreted as evidence for the spatial structure hypothesis. We have therefore removed the controlled ETT experiment from the paper. The broader findings reported in the paper rest on the larger dataset comparisons (Section 5.2), which are unaffected by this correction.
>
>
> >Kourentzes, Nikolaos, Bahman Rostami-Tabar, and Devon K. Barrow. “Demand Forecasting by Temporal Aggregation: Using Optimal or Multiple Aggregation Levels?” Journal of Business Research 78 (September 2017): 1–9. https://doi.org/10.1016/j.jbusres.2017.04.016.

---

> ### Author Response · Authors · 2026-06-28
>
> ***[Critical] Turn the regime taxonomy from descriptive into predictive. The two descriptors should be defined and ideally thresholded before looking at results, then used to predict the winner on held-out datasets that were not used to build the taxonomy. Adding more datasets per regime is essential, since "incoherent" currently contains only Traffic, and the "low-channel CI" claim rests only on the ETT family. Without out-of-sample prediction, the framework is a post-hoc relabeling of eight outcomes.***
>
>
> *Clarifying "prediction" and "quantification" in our context*
> We used the term "predict" loosely, and we admit that this was an overstatement. Our original intent was inspired by classical statistics, where data properties like stationarity or seasonality are used to guide selection between models like ARIMA and SARIMA. We intended for our spatial descriptors to serve a similar role and help practitioners "predict"  which SOFTS variant will be suitable given a dataset's spatial structure. We have revised the manuscript to make this clear
> (**lines 59-67**).
>
> Many prior works assert that inter-channel correlation drives CD model performance without actually measuring it. High-correlation fraction and block separation were proposed specifically to fill this gap and provide a reproducible, quantified terminology for reasoning about correlational structure. We concede that these descriptors have only been validated within the SOFTS variants and may not universally generalize.
>
> *On the post-hoc nature of the regime taxonomy*
> We agree. The proposed taxonomy is a classification of where C-SOFTS outperforms or degrades relative to SOFTS, derived from and validated on the same evaluation pool. We will make this explicit in the revised manuscript and present it as an exploratory characterization rather than a universally validated predictive framework. We have toned down assertions of a definitive predictive taxonomy and reframed our descriptors as diagnostic tools tailored specifically to the behavior of the SOFTS variants
> (**lines 195-196**).
>
> We still believe that they guide model selection amongst the SOFTS variants. Even in the context of C-SOFTS, this taxonomy helps practitioners determine when to use the spatial mixer.
>
> *Absence of thresholds*
> We fully intend to derive explicit thresholds in future work. However, doing so rigorously requires synthetic datasets that uniformly and independently vary HCF and BS across a dense grid of values. Using real benchmark datasets for threshold derivation would be inappropriate, as their spatial properties are not independently controlled, and 12 datasets provide insufficient coverage of the HCF–BS space to fit reliable decision boundaries.
>
> The benchmark datasets were chosen deliberately for two reasons. First, they enable direct comparison with existing literature, which is essential for demonstrating that our findings hold on the same datasets used to evaluate competing architectures. Second, real benchmarks are sufficient to establish the central empirical claim of this paper: that correlation magnitude alone does not determine the effectiveness of CD models.
>
> We have stated this as a limitation explicitly, rather than implying that the current results deliver decision boundaries
> (**lines 325-330**).

---

> > ### Author Response · Authors · 2026-06-28
> >
> > ***(2) The regime taxonomy is defined post hoc and then used to explain the same eight datasets it was derived from. The three categories (block structure, homogeneous, incoherent) are introduced after the results are seen, and "incoherent" effectively contains a single dataset (Traffic) whose definition coincides with the one dataset C-SOFTS fails on. When two datasets with nearly identical proposed descriptors disagree (PEMS03 has high-correlation fraction $1.00$, block separation $0.13$, and improves $+7.3%$; PEMS07 has $0.98$, $0.19$, and degrades $-1.71$), the explanation invokes the number of clusters (99 versus 316), which is a third quantity that is not one of the two proposed descriptors and that depends on the silhouette-based $k$ selection. So the explanatory framework keeps acquiring degrees of freedom to fit eight outcomes, which is the opposite of the "interpretable, quantified" contribution the paper claims. Notably, the paper never actually delivers thresholds on its two descriptors that predict the winner; the mapping is done by inspection.***
> >
> > *The specific utility of the descriptors*
> > We show that the proposed descriptors provide diagnostic utility to practitioners using the spatial mixer. This is best illustrated by the reviewer's points regarding the apparent contradiction between PEMS03($+7.3%$) and PEMS07($-1.71%$). Without the proposed descriptors, this outcome appears anomalous. The frequency mixer consistently improves performance across all datasets, whilst the spatial mixer is dataset dependent. It degrades performance on all benchmarks except PEMS04 and PEMS08. These two datasets have certain characteristics that seem to favour spatial compression. Their channels highly correlate with each other (HCF > 0.95), and they have a few clusters (<= 3) that are moderately distinct. Focusing on the PEMS dataset family, the ablation study shows that removing the spatial mixer degrades C-SOFTS performance on PEMS04, which has 2 clusters, by 2.7%, and on PEMS08 (3) by 0.5%. However, removing the spatial mixer improves performance on PEMS03 (99) and PEMS07 (316) by 4.9% and 12.7%, respectively.
> >
> > Combining the ablation results with the data descriptors, we suggest that the spatial mixer is beneficial only for datasets with few clusters (≤ 3) and extremely high inter-channel correlation. The cluster count should not be viewed as an additional free parameter for fitting the data, but rather as a criterion for determining when to apply the spatial mixer in high HCF conditions. These results allow practitioners using SOFTS to select the appropriate variant for a given dataset, and, when the data characteristics indicate the use of C-SOFTS, to further determine which components to include. Furthermore, these observations provide a concrete foundation for future research. Given that the spatial mixer fails when forced to compress a high number of clusters globally, a promising next step is to explore whether the spatial mixer can be applied to localized channel clusters independently.
> >
> >
> > These insights would not have been readily apparent without the proposed data descriptors.
> >
> > *Robustness of I-SOFTS*
> > The text has been revised to scope I-SOFTS performance only to the ETT dataset, and points to the fact that it shows where the STAR module might not offer additional benefits.

---

> ### Author Response · Authors · 2026-06-28
>
> ***[Critical] Justify or revise block separation and the correlation computation. Please clarify whether correlations are computed on raw or detrended series, ... since the PEMS03-versus-PEMS07 explanation hinges on the (unstable) cluster count.***
>
> *Circularity of Block Separation:*
> We select $k$ to find the most natural partition, consistent with the goal of measuring the best-case cluster contrast the data supports. A fixed $k$ would introduce arbitrary choices that would be more difficult to defend than $r=0.5$. The circularity concern would apply if we were claiming to validate cluster existence, but block separation is an exploratory descriptor, not a hypothesis test. Its goal is to quantify the correlational relationship between existing natural clusters.
>
> *Stationarity or detrending:*
> This is a valid concern given how we initially framed the descriptors, but our intention was for high-correlation fraction and block separation to characterize the data as-is, not to mirror the model's internal processing. The Augmented Dickey-Fuller (ADF) is applied to raw data to determine the suitability of the data for ARIMA, even though ARIMA might differ the data internally. HCF and BS are intended to serve the same role, which is to determine which SOFTS variant to use. They measure the global spatial dependence actually present in the observed data, regardless of whether that dependence arises from shared trends, seasonal structure, or otherwise. If all sensors in the Solar dataset rise and fall together due to the sun's position, then the dataset genuinely exhibits strong cross-channel dependence, and that is exactly what we want to capture.
>
>
> *Arbitrary selection of high-correlation fraction threshold:*
> The threshold $r=0.5$ follows the conventional interpretation of Pearson correlation magnitude, where values above 0.5 are considered moderately to strongly correlated. We acknowledge that it is a choice.  To assess sensitivity, we recomputed HCF across thresholds $r \in [0,1]$ for all datasets. The first table shows the dataset ranking for each threshold, whilst the second table presents the HCF scores underlying these rankings.
>
> |   Rank | r=0.0       | r=0.1       | r=0.2       | r=0.3       | r=0.4       | r=0.5       | r=0.6       | r=0.7       | r=0.8       | r=0.9       |
> |-------:|:------------|:------------|:------------|:------------|:------------|:------------|:------------|:------------|:------------|:------------|
> |      1 | PEMS03      | PEMS03      | PEMS03      | PEMS03      | PEMS03      | Solar       | Solar       | Solar       | Solar       | Solar       |
> |      2 | PEMS04      | PEMS07      | Solar       | Solar       | Solar       | PEMS03      | PEMS03      | PEMS03      | PEMS03      | PEMS03      |
> |      3 | PEMS07      | PEMS08      | PEMS07      | PEMS07      | PEMS07      | PEMS07      | PEMS07      | PEMS07      | PEMS07      | PEMS08      |
> |      4 | PEMS08      | Solar       | PEMS04      | PEMS04      | PEMS04      | PEMS08      | PEMS08      | PEMS08      | PEMS08      | PEMS07      |
> |      5 | ECL | PEMS04      | PEMS08      | PEMS08      | PEMS08      | PEMS04      | PEMS04      | PEMS04      | PEMS04      | PEMS04      |
> |      6 | Solar       | Traffic     | Traffic     | Traffic     | Traffic     | Traffic     | Traffic     | ECL | ECL | Weather     |
> |      7 | Traffic     | ECL | ECL | ECL | ECL | ECL | ECL | Traffic     | weather     | ECL |
> |      8 | Weather     | Weather     | Weather     | Weather     | Weather     | Weather     | Weather     | Weather     | Traffic     | Traffic     |
>
> Whilst increasing $r$ reduces the HCF values, the relative ordering of datasets is stable across all reasonable thresholds. Since our conclusions, to a greater extent, rest on this ordering rather than on absolute HCF values, they are not an artifact of the specific threshold chosen. Highly uniform datasets like Solar and the PEMS variants consistently rank higher, whilst ECL, Traffic, and Weather rank lower.
>
> ---
> ***[Critical] Fix the factual inconsistencies. (i) The dataset table (Table 6) lists "PEMS05/PEMS06" with 883/170 channels, but the paper everywhere else uses "PEMS07/PEMS08" for those same channel counts; reconcile the naming and confirm exactly which datasets were used. (ii) Correct "highest MSE" to "lowest MSE" (best) in Sections 5.1 and 5.2. (iii) Confirm the "twelve datasets" count is consistent with what is actually evaluated.***
>
> *PEMS05/PEMS06 vs PEMS07/PEMS08*
> This error is an MS Excel artefact and has been duly corrected to the appropriate PEMS07/PEMS08.
>
> *MSE Wording*
> The text has been revised to properly state that the I-SOFTS and C-SOFTS had the *lowest* MSEs.
>
> *Twelve Dataset Count*
> We would like to confirm that we evaluated 12 datasets: 4 PEMS, 4 ETT, 1 Traffic, 1 Solar, 1 Weather, and 1 ECL.

---

> > ### Author Response · Authors · 2026-06-28
> >
> > ***[Strengthen] Make the multi-model tables a controlled comparison, or label them clearly. Since the iTransformer, PatchTST, and other numbers are taken from the SOFTS paper rather than re-run, state this prominently and avoid claims like "best on 6/8 datasets" that imply a controlled sweep. The variant-versus-SOFTS comparison is the rigorous one, so lead with it.***
> >
> > We agree that the SOFTS-versus-variant comparison is the primary controlled finding. The multi-model tables are retained for contextual reference, consistent with standard practice in the field. We have clarified in the paper that the average dataset-level performances are relative to only the SOFTS models. Table captions explicitly state that only SOFTS results were reproduced.
> >
> >
> > ---
> >
> > ***[Strengthen] Broaden beyond a single lookback and backbone. Results use a fixed lookback $L=96$ and a single model family (SOFTS). At least one additional lookback, and ideally a second hybrid or CD backbone, would tell us whether the regime effect is a property of the data or of the STAR plus Channel-Mixer design specifically. The paper's own generality claims require this.***
> >
> > We acknowledge that our initial manuscript overstated the generality of our results. Our empirical results are specific to the SOFTS variants, and the text has been revised to reflect this scope.
> >
> > ---
> >
> >
> > ***[Strengthen] Temper the abstract. "Up to $15.9$" is a single best case (PEMS04); reporting the average and the spread would be more honest. Please also soften the "primary determinant" language to match the currently correlational, small-sample evidence.***
> >
> > The abstract has been updated to simply state the identified conditions under which C-SOFTS outperforms S-SOFTS.